# DOBF: A Deobfuscation Pre-Training Objective for Programming Languages

**Marie-Anne Lachaux**[*]
Facebook AI Research
malachaux@fb.com

**Baptiste Roziere**[*]
Facebook AI Research
Paris-Dauphine University
broz@fb.com

**Marc Szafraniec**
Facebook AI Research
szafraniec@fb.com

**Guillaume Lample**
Facebook AI Research
glample@fb.com

## Abstract

Recent advances in self-supervised learning have dramatically improved the state of the art on a wide variety of tasks. However, research in language model pre-training has mostly focused on natural languages, and it is unclear whether models like BERT and its variants provide the best pre-training when applied to other modalities, such as source code. In this paper, we introduce a new pre-training objective, DOBF, that leverages the structural aspect of programming languages and pre-trains a model to recover the original version of obfuscated source code. We show that models pre-trained with DOBF significantly outperform existing approaches on multiple downstream tasks, providing relative improvements of up to 12.2% in unsupervised code translation, and 5.3% in natural language code search. Incidentally, we found that our pre-trained model is able to deobfuscate fully obfuscated source files, and to suggest descriptive variable names.

## 1 Introduction

Model pre-training with self-supervised methods such as BERT Devlin et al. [2018], RoBERTa Liu et al. [2019], XLM Lample and Conneau [2019] or XLNet Yang et al. [2019], has become ubiquitous in Natural Language Processing (NLP), and led to significant improvements in many tasks. These approaches are based on the Masked Language Modeling (MLM) objective, which consists in randomly masking words from an input text, and training a model to recover the original input. In the original approach proposed by Devlin et al. [2018], a fraction of selected masked words is replaced by masked tokens, another is replaced by random words, and another remains unchanged. Since then, a myriad of studies have proposed to modify the MLM objective, either by masking contiguous spans of text Song et al. [2019], Joshi et al. [2020], masking named entities and phrases Sun et al. [2019], sampling masked words according to their frequencies Lample and Conneau [2019], replacing words with plausible alternatives Clark et al. [2020], etc. Overall, most of these pre-training objectives boil down to denoising auto-encoding tasks with different methods to add noise to the input, using arbitrary noise functions. In our case, we are interested in pre-training deep learning models for programming languages. As in natural language, pre-training was shown to be effective for source code Feng et al. [2020], Roziere et al. [2020]. However, these studies both rely on the original MLM objective proposed by Devlin et al. [2018], which was initially designed for natural languages and does not leverage the particular structure of source code. We argue that this objective

---

[*]Equal contribution. The order was determined randomly.

35th Conference on Neural Information Processing Systems (NeurIPS 2021).

is actually suboptimal in the context of programming languages, and propose a new objective based on deobfuscation of identifier names in source code.

Code obfuscation consists in modifying source code in order to make it harder for humans to understand, or smaller while keeping its behaviour unchanged. In some ancient interpreted languages, name minimization could also reduce the memory usage of the program. Today, it is used to protect intellectual property by preventing people from understanding and modifying the code, to prevent malware detection, and to compress programs (e.g. Javascript code) to reduce network payload sizes. Moreover, C compilers discard variable names, and current rule-based and neural-based decompilers generate obfuscated C code with uninformative variable names Fu et al. [2019]. Obfuscators typically apply several transformations to the code. While some operations can be reversed (e.g. dead code injection), the obfuscation of identifier names—renaming every variable, method and class with uninformative names—is irreversible and has a substantial impact on code comprehension Gellenbeck and Cook [1991], Takang et al. [1996], Lawrie et al. [2006].

By analyzing the overall structure of an obfuscated file, an experienced programmer can always, with time, understand the meaning of the obfuscated code. For instance, in the obfuscated example in Figure 1, one can recognize the function and guess that it implements a breadth-first search algorithm. We also expect neural networks, that excel in pattern recognition, to perform well on this task. We propose to pre-train a model to revert the obfuscation function, by training a sequence-to-sequence (seq2seq) model to convert obfuscated functions, where names of functions and variables have been replaced by uninformative names, back to their original forms. Suggesting proper variable and function names is a difficult task that requires to understand what the program does. In the context of source code, it is a more sensible, but also a more difficult task than MLM. Indeed, we observe (c.f. Figure 1) that predicting the content of randomly masked tokens is usually quite simple, as it often boils down to making syntax related predictions (e.g. predicting that was has been masked out is a parenthesis, a semi-column, etc.). These simple predictions actually provide little training signal to the model. In practice, MLM also masks out variable names, but if a given variable appears multiple times in a function, it will be easy for the model to simply copy its name from one of the other occurrences. Our model does not have this issue, as all occurrences of masked variables are replaced by the same `VAR_i` special tokens.

In this paper, we make the following contributions:

- We present DOBF, a new pre-training objective based on deobfuscation, and show its effectiveness on multiple programming languages.
- We show that DOBF significantly outperform MLM (e.g. BERT) on multiple tasks such as code search, code summarization or unsupervised code translation. We show that pre-training methods based on DOBF outperform all existing pre-training methods on all the considered tasks.
- We show that, by design, models pre-trained with DOBF have interesting applications and can be used to understand functions with uninformative identifier names. Besides, the model is able to successfully deobfuscate fully obfuscated source files.

Our method improves other machine learning methods for programming languages. Automatic deobfuscation and identifier name proposal can also make code more accessible, and facilitate innovation and malware detection. Conversely, automatic deobfuscation could facilitate theft of proprietary code, therefore hindering the distribution of software and reducing investments in innovative softwares. Socially undesirable uses of our work (e.g. intellectual property theft) are targetable legally, while desirable ones (e.g. malware detection, IDE tools) can be seen as primarily technical problems. Therefore, we believe that the impact of our work will be mostly positive.

## 2 Related work

**Masked Language Modeling pre-training.** Large pre-trained transformers such as BERT Devlin et al. [2018] or RoBERTa Liu et al. [2019] led to significant improvements in the majority of natural language processing tasks. The quality of pre-training mainly comes from the MLM objective (i.e. the cloze task), that allows the model to make predictions by leveraging left and right contexts, unlike causal language modeling (CLM) where the model predictions are only conditioned on previous words. In MLM, the model takes as input a sentence and uniformly selects 15% of its tokens. Of the

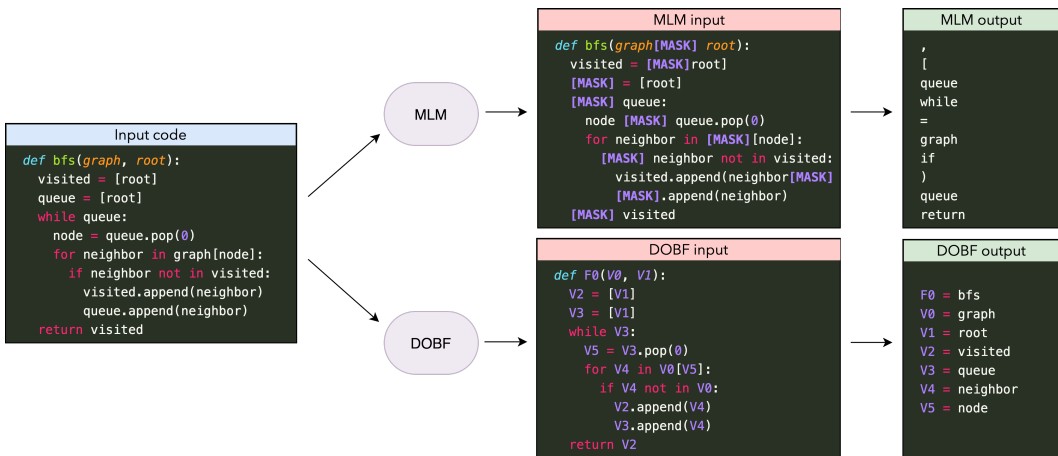

Figure 1: **Illustration of the MLM and DOBF objectives.** Given an input function, the masked language modeling (MLM) task randomly samples tokens to mask out. With source code, a large fraction of these tokens are related to the language syntax (e.g. commas, parentheses, etc.) that are trivial for the model to predict, and provide a poor training signal. Instead, we propose to obfuscate the code by masking the name of functions and variables, and to train the model to recover the original function by deobfuscating the code (DOBF). When a variable is masked out, we mask all occurrences of this variable with the same mask symbol (e.g. all occurrences of "visited" are replaced by "V0") to prevent the model from copying names. The DOBF objective is more difficult and provides a better learning signal.

selected tokens, 80% are replaced by a special symbol [MASK], 10% are left unchanged, and the remaining 10% are replaced by random tokens from the vocabulary. The MLM objective consists in recovering the initial sentence given the corrupted one. Lample and Conneau [2019] noticed that the masked words are often easy to predict, and proposed to sample the 15% masked words according to their frequencies instead of uniformly. This way, rare words are sampled more often, making the pre-training task more difficult for the model, which results in a better learning signal and faster training. Sun et al. [2019] also noticed that recovering the tokens masked by MLM is too simple in some contexts (e.g. predicting the two tokens "Harry Potter" is much harder than predicting only "Harry" if you know the next word is "Potter"). To address this issue, they proposed to mask phrases and named entities instead of individual tokens. Joshi et al. [2020] and Song et al. [2019] made a similar observation and proposed to mask random spans of text. They showed that this simple modification improves the performance on many downstream NLP tasks.

**Alternative objectives.** Other pre-training objectives have been proposed in addition to MLM. For instance, Devlin et al. [2018] also uses the next sentence prediction (NSP) objective, a binary classification task that consists in predicting whether two input sentences follow each other in the original corpus. The NSP objective was originally designed to improve the performance on downstream NLP tasks, but recent studies Lample and Conneau [2019], Liu et al. [2019] showed that training MLM on a stream of sentences to leverage longer context, and removing the NSP objective improves the quality of pre-training. To improve the sample-efficiency of MLM (where only 15% of tokens are predicted), Electra Clark et al. [2020] proposed to replace (and not mask) some tokens with plausible alternatives, and to train a network to detect the tokens that have been replaced. They showed that this new Replaced Token Detection (RTD) objective matches the performance of RoBERTa while using four times less computational resources. Dong et al. [2019] proposed a model that combines multiple pre-training tasks, including bidirectional, but also left-to-right and right-to-left language modeling objectives. Lewis et al. [2019] also proposed different pre-training objectives, to detect whether input sentences have been permuted, tokens have been deleted or inserted, etc.

**Code Generation Pre-training.** Recent studies showed that pre-training methods developed for natural language processing are also effective for programming languages. For instance, Feng et al. [2020] proposed CodeBERT, a RoBERTa-based model trained on source code using the MLM and RTD objectives. With GraphCodeBERT Guo et al. [2020], the MLM objective is complemented by an edge-prediction objective, in which the model predicts edges in the data flow graph to make the

model understand the structure of the code. In Jain et al. [2020], a model is trained on javascript code using a contrastive loss ensuring that the representations are robust to some semantic-preserving transformations. They showed that their model performs well on downstream code generation tasks and outperforms previous pre-training approaches. Kanade et al. [2020] applied MLM and the next sentence prediction objectives to pre-train models on Python code. More recently, Roziere et al. [2020] applied the unsupervised machine translation principles of Lample et al. [2018a,b] to monolingual source code from GitHub. They showed that the resulting model, TransCoder, was able to translate source code between Python, Java, and C++, in a fully unsupervised way. In this paper, we propose to use a code-specific objective to better pre-train models designed to be fine-tuned on code generation tasks: code deobfuscation. Machine learning is frequently used on tasks involving programming languages, including code completion Li et al. [2018], Liu et al. [2020], Kim et al. [2020], Svyatkovskoy et al. [2020], bug detection and code repair Allamanis et al. [2018], Wang et al. [2017], Chen et al. [2019], Murali et al. [2020], Tufano et al. [2019], Tarlow et al. [2020], code summarization Alon et al. [2019a], Hu et al. [2018], Xie et al. [2021], clone detection Wei and Li [2017], Ain et al. [2019], Wang et al. [2020], code search Gu et al. [2018], Cambronero et al. [2019] and code translation Chen et al. [2018], Roziere et al. [2020]. Most of these tasks can benefit from pre-trained models that capture the semantics of the code.

**Code deobfuscation.** Empirical studies show that naming conventions and the use of informative identifier names make code more understandable, easier to maintain and lead to fewer bugs Takang et al. [1996], Liblit et al. [2006], Butler et al. [2009]. It motivated other works studying deobfuscation of identifier names and identifier name proposal using n-grams Allamanis et al. [2014, 2015], probabilistic models Raychev et al. [2015], Bichsel et al. [2016], Vasilescu et al. [2017], Alon et al. [2018], and recurrent neural networks Bavishi et al. [2018], Lacomis et al. [2019]. Alon et al. [2018] extract features from Abstract Syntax Tree (AST) paths and train a Conditional Random Field to predict variable and method names, and infer types for several languages. DIRE Lacomis et al. [2019] uses a commercial decompiler to obtain C code with uninformative identifier names from binaries. They also use AST features, which go through a Graph Neural Network trained jointly with a LSTM model on the sequence of C tokens to retrieve relevant identifier names. More recently, David et al. [2020] used a transformer together with augmented representations obtained from static analysis to infer procedure names in stripped binary files. These models are already used to understand obfuscated and compiled source code. However, none of these studies investigated the use of deobfuscation for model pre-training.

## 3 Model

### 3.1 MLM and denoising for Programming Languages

A countless number of pre-training objectives have been introduced in the literature Devlin et al. [2018], Clark et al. [2020], Lewis et al. [2019], Liu et al. [2019], Dong et al. [2019]. Most of them rely on hyper-parameters and seemingly arbitrary decisions (Should we mask individual tokens or spans? Which fraction of them? What do we do with masked out tokens? etc.). These choices are typically based on intuition and validated empirically on natural language processing tasks. However, source code is much more structured than natural language, which makes predicting masked tokens much easier for programming languages.

The first row in Figure 1 shows an example of input / output for the MLM objective. We can see that the majority of tokens are composed of Python keywords or symbols related to syntax: `,` `[` `while` `=` `if` `)` `return`. These symbols are easy to recover, and a model will quickly learn to predict them with perfect accuracy. This effect is accentuated by the verbosity of the language. For instance, we would see significantly more of these tokens in Java. Retrieving the obfuscated `graph` token is also relatively simple: the model only needs to retrieve the most relevant variable in the scope. More generally, retrieving an identifier name is often easy when given its full context, including its definition and usages. The denoising-auto-encoding (DAE) objective Vincent et al. [2008], which trains an encoder-decoder model to retrieve masked token and recover randomly modified input sentences, is quite similar to MLM and the model can also retrieve identifier names easily by finding their definition or usages. Overall, we suspect that the MLM objective is too simple in programming languages and we introduce a new objective, DOBF, which encourages the model to learn a deeper understanding of code semantics.

## 3.2 Deobfuscation Objective

Instead of MLM, we propose a new pre-training objective, DOBF, that leverages the particular structure of programming languages. We obfuscate code snippets by replacing class, function and variable names with special tokens, and train a model to recover the original names. When an identifier is selected, all of its instances in the code are replaced by the same special token. This differs from MLM where the name of a variable can appear multiple times while being masked a single time. For instance, in Figure 1, DOBF will replace the two occurrences of `node` by the same symbol V5, while MLM will only mask one of these occurrences. As a result, the fraction of meaningful tokens masked by the objective is language independent: for more verbose languages (e.g. Java), the less informative syntax-related tokens will not be masked out by the DOBF objective.

Each identifier is replaced with probability $p_{obf} \in [0, 1]$. We ensure that the original input is modified: if no identifier is replaced, we draw a random one to obfuscate. When $p_{obf} = 0$, we always obfuscate exactly one random identifier in the input. When $p_{obf} = 1$, we obfuscate all the identifiers defined in the file. We ensure that the obfuscated code has the same behavior as the original. The second row in Figure 1 shows an example of obfuscated code with $p_{obf} = 1$, where we obfuscate a function `bfs` which implements a breadth-first search. The function `append` is not obfuscated as it is a standard Python function not defined in the file. The model is given the obfuscated code as input and has to restore the original name of each special token `CLASS_i`, `FUNC_i` and `VAR_i`. In other words, the model needs to output a dictionary mapping special tokens to their initial values.

Finding informative names for obfuscated identifiers requires the model to learn a deep understanding of code semantics, which is desirable for a pre-training task. MLM will mask only some of the occurrences of the identifiers and leave the other ones unchanged so that the model can simply copy identifier names. In Figure 1, with MLM masking, the model can simply notice that a variable named `queue` is called on the fourth line. Since the variable is not defined, the model can easily guess that `queue` has to be defined on the third line, and infer the value of the corresponding `[MASK]` token. With the deobfuscation objective, the model needs to analyze code patterns and understand the semantics of the variable to infer that, since its elements are popped with `.pop(0)`, the variable V3 implements a queue. If its elements were popped with `.pop()`, our model would name it `stack` instead of `queue` (c.f. Figure 7 in the appendix).

## 3.3 Implementation

Overall, the deobfuscation objective operates like a supervised machine translation objective, where a seq2seq model is trained to map an obfuscated code into a dictionary represented as a sequence of tokens. At inference time, the model is able to suggest meaningful class, function and variable names for a piece of code with an arbitrary number of obfuscated identifiers. Obfuscated classes, functions, and variables, are replaced with associated special tokens: `CLASS_0 ... CLASS_N`, `FUNC_0 ... FUNC_N` and `VAR_0 ... VAR_N`. We serialize the output dictionary as a sequence of tokens where the entries are separated by a delimiter symbol `|`. [2]

# 4 Experiments

We train DOBF with the deobfuscation objective. First, we evaluate our model on two straightforward deobfuscation applications. Then, we show its performance on multiple downstream tasks.

## 4.1 Deobfuscation

We evaluate our model on two applications of the deobfuscation task: when $p_{obf} = 0$ (the model has to retrieve a single identifier name), and $p_{obf} = 1$ (the model has to retrieve all the identifier names).

**Deobfuscating a single identifier**   When $p_{obf} = 0$, only one identifier is obfuscated. In that case, the model has to propose a relevant name for that identifier using the rest of the non-obfuscated file as context. It can be used as a tool that suggests relevant variable names. Integrated development environments (e.g. PyCharm, VSCode) already perform this task, often using handcrafted rules.

---

[2]In the obfuscated example given in Figure 1, the model is trained to generate: `FUNC_0 bfs | VAR_0 graph | VAR_1 root | VAR_2 visited | VAR_3 queue | VAR_4 neighbor | VAR_5 node`.

**Deobfuscating all identifiers** Obfuscators are commonly used to make code smaller and more efficient or to protect it by making it more difficult to understand and reuse. They typically apply several transformations, one of them being to replace every identifier name with short and uninformative names (e.g. a, b, c). In our work, such a transformation corresponds to obfuscating a file with $p_{obf} = 1$. To measure our model's ability to revert the obfuscation operation, we evaluate its accuracy when obfuscating all identifier names. Another application would be to help understand source code written with uninformative variable names.

**Evaluation metric** We evaluate the ability of our model to retrieve identifier names from the original non-obfuscated code. We report the accuracy, which is the percentage of recovered tokens that exactly match the ground truth. Following previous works Allamanis et al. [2015, 2016], Alon et al. [2018, 2019b], we also report the *subtoken score*, a more flexible metric which computes the precision, recall, and F1 scores for retrieving the original case-insensitive subtokens. Each token is broken into subtokens using uppercase letters for camlCase and underscores for snake_case. For instance, `decoderAttention` would be considered to be a perfect match for `decoder_attention` or `attentionDecoder`. `attention` would have a perfect precision but a recall of $0.5$, so a F1 score of 66.7. `crossAttentionDecoder` would have a perfect recall but a precision of $\frac{2}{3}$, corresponding to a F1 score of $80.0$. We compute the overall subtoken precision, recall and F1 scores averaged over each file in our validation and test datasets.

## 4.2 Fine-tuning on downstream tasks

In order to evaluate DOBF as a pre-training model, we fine-tune DOBF on TransCoder and on three tasks from CodeXGLUE Lu et al. [2021], a benchmark for programming languages. The data, code and models from CodeXGLUE and TransCoder are available respectively under the MIT and the Creative Commons license. We only consider the Java and Python tasks with an encoder in the model architecture for which the training, validation, and test sets are publicly available.

**CodeXGLUE Clone Detection** This task is a binary classification problem where the model has to predict whether two code snippets are semantically equivalent. It is evaluated using the macro F1 score. The model is composed of a single encoder and a classification layer. An input consists in two snippets of code, which are concatenated before being fed to the model. This task is available in Java.

**CodeXGLUE Code Summarization** Given a code snippet, the model is trained to generate the corresponding documentation in natural language. The architecture is a sequence-to-sequence transformer model evaluated using BLEU score Papineni et al. [2002]. The dataset includes both Java and Python source code.

**CodeXGLUE NL Code Search** Given a code search query in natural language the model has to retrieve the most semantically related code within a collection of code snippets. This is a ranking problem evaluated using the Mean Reciprocal Rank (MRR) metric. The model is composed of two encoders. The natural language query and the code are encoded separately, and we compute the dot product between the first hidden states of the encoders' last layers. This task is available in Python.

**TransCoder** TransCoder Roziere et al. [2020] is an unsupervised machine translation model which translates functions and methods between C++, Java, and Python. A single seq2seq model is trained for all languages. In the original work, TransCoder is pre-trained with MLM, and trained with denoising auto-encoding and back-translation. TransCoder is evaluated using the Computational Accuracy metric, which computes the percentage of correct solutions according to series of unit tests. We only consider a single model output (CA@1), with beam sizes of 1 and 10.

## 4.3 Experimental details

**Model Architecture** We consider a seq2seq model with attention, composed of an encoder and a decoder using a transformer architecture Vaswani et al. [2017]. We train models with the same architecture and tokenizer as CodeBERT Feng et al. [2020] and GraphCodeBERT Guo et al. [2020] in order to provide fair comparisons: 12 layers, 12 attention heads and a hidden dimension of 768. We also train a model with the same parameters as TransCoder (see Figure 4 in the Appendix).

**Training dataset** As in Roziere et al. [2020], we use the GitHub public dataset available on Google BigQuery and select all Python and Java files within the projects with licenses authorizing use for

```
def FUNC_0(VAR_0, VAR_1):                    def bfs(graph, start):
    VAR_2 = [VAR_1]                              visited = [start]
    VAR_3 = [VAR_1]                              queue = [start]
    while VAR_3:                                 while queue:
        VAR_4 = VAR_3.pop(0)                         node = queue.pop(0)
        for VAR_5 in VAR_0[VAR_4]:                   for neighbor in graph[node]:
            if (VAR_5 not in VAR_2):                     if (neighbor not in visited):
                VAR_2.add(VAR_5)                             visited.add(neighbor)
                VAR_3.append(VAR_5)                          queue.append(neighbor)
    return VAR_2                                 return visited
```

Figure 2: **Full deobfuscation of a breadth-first-search function by DOBF.** The code on top has been fully obfuscated. The code on the bottom was recovered using DOBF by replacing the function name and every variable name using the generated dictionary. DOBF is able to suggest relevant function and variable names. It makes the code much more readable and easier to understand.

research purposes. Following Lopes et al. [2017] and Allamanis [2019], we remove duplicate files. We also ensure that each fork belongs to the same split as its source repository. We obfuscate each file and create the corresponding dictionary of masked identifier names, resulting in a parallel (obfuscated file - dictionary) dataset of 19 GB for Python and 26 GB for Java. We show some statistics about this dataset in Table 3 in the appendix. For comparison purposes, we apply either the BPE codes used by Roziere et al. [2020] or by Feng et al. [2020]. In practice, we train only on files containing less than 2000 tokens, which corresponds to more than 90% and 80% of the Java and Python files respectively.

**Training details** We train DOBF to translate obfuscated files into lists of identifier names. During DOBF training, we alternate between batches of Java and Python composed of 3000 tokens per GPU. We optimize DOBF with the Adam optimizer Kingma and Ba [2014] and an inverse square-root learning rate scheduler Vaswani et al. [2017]. We implement our models in PyTorch Paszke et al. [2019] and train them on 32 V100 GPUs for eight days. We use float16 operations to speed up training and to reduce the memory usage of our models. We try different initialization schemes: training from scratch and with a Python-Java MLM model following Roziere et al. [2020]. We train DOBF with three different obfuscation probability parameters: $p_{obf} \in \{0, 0.5, 1\}$. For each $p_{obf}$ value, we train models with multiple initial learning rates ranging from $10^{-4}$ to $3.10^{-4}$ and select the best one using the average subtoken F1 score computed on the validation dataset.

**Fine-tuning details** Depending on the fine-tuning tasks, we consider different model architectures: seq2seq models with encoder and decoder, architectures with two encoders or a single encoder. In all cases, we initialize the encoders of these models with the encoder of DOBF and fine-tune all parameters. For fair comparison, we rerun all baselines, and train models with the same architectures, number of GPUs, batch sizes and optimizers. For CodeXGLUE, we noticed that the tasks are quite sensitive to the learning rate parameter used during fine-tuning. We perform a grid search on five learning rate parameters ranging from $5.10^{-6}$ to $10^{-4}$ and we select the best parameter on the validation dataset. For TransCoder, we use a learning rate of $10^{-4}$ as in Roziere et al. [2020] and we train the models for 2 day on 32 Tesla V100 GPUs.

## 5 Results

### 5.1 Deobfuscation

In Table 1, we evaluate the ability of our model to recover identifier names, either when only one identifier is obfuscated ($p_{obf} = 0$) or when all identifiers are obfuscated ($p_{obf} = 1$), for models trained with $p_{obf} \in \{0, 0.5, 1\}$. Even when evaluating with $p_{obf} = 0$, training with $p_{obf} = 0$ is less efficient than $p_{obf} = 0.5$ since the model is only trained to generate a single variable for each input sequence. Training with $p_{obf} = 0.5$ is a more difficult task that requires the model to learn and understand more about code semantics. Forcing the model to understand the structure of the code may be useful even when testing with $p_{obf} = 0$, as some identifier names cannot be guessed only from the names of other identifiers. When DOBF has to recover a fully obfuscated function, it obtains the best accuracy when trained with $p_{obf} = 1$. It manages to recover 45.6% of the initial identifier names. We also observe that, for every configuration, initializing DOBF with MLM improves the performance.

Figure 2 shows an example of a fully obfuscated function recovered by our model. DOBF successfully manages to understand the purpose of the function and to predict appropriate variable names. Figure 3 shows examples of function name proposal by DOBF for functions implementing matrix operations in

| Input Code | Function Name Proposals | |
|---|---|---|

```python
def FUNC_0 (m1, m2):
  assert m1.shape == m2.shape
  n, m = m1.shape
  res = [[0 for _ in range(m)] for _ in range(n)]
  for i in range(n):
    for j in range(m):
      res[i][j] = m1[i][j] + m2[i][j]
  return res
```
matrix_add    25.9%
matrixAdd    22.5%
matrixadd    18.8%
matrix_sum    16.7%
matrix_addition    16.1%

```python
def FUNC_0 (matrix):
  n, _ = matrix.shape
  for i in range(n):
    for j in range(i,n):
      matrix[i][j], matrix[j][i] = \
        matrix[j][i], matrix[i][j]
```
transpose    36.7%
rotate    29.5%
rotate_matrix    17.1%
symmetric    8.9%
rotate_matrix_by_row    7.7%

```python
def FUNC_0 (m1, m2):
  n1, m1 = m1.shape
  n2, m2 = m2.shape
  assert n2 == m1
  res = [[0 for _ in range(m2)] for _ in range(n1)]
  for i in range(n1):
    for j in range(m2):
      res[i][j] = sum([m1[i][k] * m2[k][j]
                    for k in range(n2)])
  return res
```
matrix_product    28.8%
mat_mult    23.8%
matmul_mat    17.0%
matprod    16.0%
matrixProduct    14.4%

Figure 3: **Additional examples of function name proposals for matrix operations in Python.** DOBF is able to find the right name for each matrix operation, showing that it learned to attend to the most important parts of the code. Even when the functions are similar, DOBF successfully and confidently (c.f. scores) understands the semantics of the function and its purpose.

Table 1: **Results on partial and full deobfuscation.** Token accuracy and subtoken F1 score of DOBF evaluated with $p_{obf} = 0$ (i.e. name proposal, where a single token is obfuscated) and $p_{obf} = 1$ (i.e. full deobfuscation, where all tokens are obfuscated). We consider models trained with different obfuscation probabilities $p_{obf}$. $DOBF_{0.5}$ performs well for both tasks, and it even performs better than $DOBF_0$ for Identifier Name Proposal. $DOBF_0$ and $DOBF_1$ perform poorly when evaluated on other $p_{obf}$ parameters. Pre-training DOBF with MLM further improves the performance.

| | Eval $p_{\text{obf}} = 0$ | | Eval $p_{\text{obf}} = 1$ | |
|---|---|---|---|---|
| | Acc | F1 | Acc | F1 |
| $DOBF_0$ | 56.3 | 68.0 | 0.4 | 0.9 |
| $DOBF_{0.5}$ | 61.1 | 71.2 | 41.8 | 54.8 |
| $DOBF_1$ | 18.1 | 27.0 | 45.6 | 58.1 |
| $DOBF_{0.5}$ init MLM | **67.6** | **76.3** | 45.7 | 58.0 |
| $DOBF_1$ init MLM | 20.0 | 28.3 | **49.7** | **61.1** |

Python. We observe that DOBF manages to identify the key tokens and to properly infer the purpose of similar but very different functions. Figures 4, 5, and 6 in the appendix show additional examples of function name proposals by DOBF in Java and Python. Figure 7 in the appendix shows additional examples where we show that DOBF also leverages non-obfuscated identifier names to understand the meaning of input functions. Figures 8 and 9 in the appendix show examples of deobfuscation of fully obfuscated Python code snippets using DOBF. It is able to understand the semantics and purposes of a variety of obfuscated classes and functions, including a LSTM cell.

## 5.2 Downstream tasks

Our results on downstream task using the same architecture as CodeBERT and GraphCodeBERT are shown in Table 2 and discussed below. Our results using the architecture of TransCoder are shown on Table 4 in the Appendix. For fine-tuning, we considered models pre-trained with $p_{obf} = 0.5$

Table 2: **Results on downstream tasks for different pre-training configurations.** Models pre-trained with DOBF initialized with MLM significantly outperform both CodeBERT and models trained with MLM only. DOBF+DAE outperforms other models on every task but clone detection, on which CodeBERT scores much higher than our MLM. It outperforms GraphCodeBERT by 0.02 MRR (+5.3%) on natural language code search (NLCS), and by 4.6% in Java → Python computational accuracy with beam size 10 (+12.2% correct translations). The tasks where MLM provides large improvements over the transformer baseline (first row, no pre-training) are also the tasks where DOBF provides the largest gains (clone detection, NL code search, unsupervised translation). The DAE baseline (initialized with MLM) already provides substantial improvements over MLM on most tasks and yields the best results for Python to Java translation while its results are poor for Java to Python.

| | Clone Det (F1 score) | Code Sum Java (BLEU) | Code Sum Python (BLEU) | NLCS (MRR) | Python→Java (CA@1) | | Java→Python (CA@1) | |
| --- | --- | --- | --- | --- | --- | --- | --- | --- |
| | | | | | k=1 | k=10 | k=1 | k=10 |
| Transformer | 88.14 | 16.58 | 16.43 | 0.025 | 24.0 | 28.4 | 29.0 | 29.7 |
| MLM | 91.89 | 18.59 | 17.95 | 0.308 | 44.8 | 45.4 | 34.5 | 35.6 |
| DAE | 96.30 | 19.19 | 18.28 | 0.380 | **48.3** | **49.2** | 32.1 | 32.8 |
| CodeBERT | 96.50 | 18.25 | 18.22 | 0.315 | 40.8 | 45.6 | 36.5 | 36.7 |
| GraphCodeBERT | 96.38 | 18.78 | 18.51 | 0.377 | 44.3 | 44.1 | 35.6 | 37.8 |
| DOBF init scratch | **96.52** | 18.19 | 17.51 | 0.272 | 43.9 | 44.1 | 35.2 | 34.7 |
| DOBF | 95.87 | 19.05 | 18.24 | 0.383 | 43.5 | 44.1 | 38.7 | 40.0 |
| DOBF+DAE | 95.82 | **19.36** | **18.58** | **0.397** | 46.6 | 47.3 | **40.6** | **42.4** |

and $p_{obf} = 1$. Since they gave very similar results on downstream tasks, we only use models pre-trained with $p_{obf} = 0.5$ in the rest of the paper. We initialize DOBF with MLM as it leads to better performance on our deobfuscation metrics. We still consider DOBF initialized randomly as a baseline in Table 2. We also consider a version where DOBF is trained together with a denoising auto-encoding (DAE) objective Vincent et al. [2008], which was shown to be effective at learning code representations in Roziere et al. [2020]. With DAE, the model is trained to recover the original version of a sequence which has been corrupted (by removing and shuffling tokens). As baselines, we consider a randomly initialized model and a model pre-trained with MLM only, and a model pre-trained with denoising and initialized with MLM. For CodeXGLUE tasks, we also consider CodeBERT as a baseline. We compare results for DOBF trained from scratch and DOBF initialized with MLM, and report results in Table 2. The randomly initialized model is useful to measure the importance of pre-training on a given task. Pre-training is particularly important for the NLCS task: without pre-training, the model achieves a performance of 0.025 MMR while it goes up to 0.308 with MLM pre-training. The main differences between our MLM baseline and CodeBERT, are that 1) CodeBERT was trained on a different dataset which contains functions with their documentation, 2) it uses an additional RTD objective, and 3) is initialized from a RoBERTa model. Although code summarization and NL code search involve natural language and may benefit from CodeBERT's dataset that contains code documentation, we obtained very similar results on this task using a simpler dataset. However, our MLM baseline did not match their performance on clone detection. We also tried to initialize our MLM model with RoBERTa, but did not observe any substantial impact on the performance on downstream tasks.

The models based on DOBF obtain state-of-the-art results on all downstream tasks, outperforming GraphCodeBERT, CodeBERT and MLM. The deobfuscation objective is already effective as a pre-training task. Even when initialized randomly, it leads to results comparable to MLM on most tasks and is much more effective on clone detection. The DOBF+DAE model outperforms MLM on all downstream tasks, the major improvement being for NL code search, which is also the task that benefited the most from MLM pretraining For unsupervised translation, DOBF+DAE increases the computational accuracy by 1.9% when translating from Python to Java, and by 6.8% when translating from Java to Python with beam size 10. Also, DOBF beats CodeBERT by a wide margin on NL code search and code summarization, showing that programming language data aligned with natural language is not necessary to train an effective model on those tasks. DOBF initialized with MLM and combined with DAE yields higher scores than both DOBF alone and MLM, on most tasks. It shows that objectives such as MLM and DAE that provide unstructured noise are complementary to DOBF.

# 6   Conclusion

In this paper, we introduce a new deobfuscation objective and show that it can be used for three purposes: recover fully obfuscated code, suggest relevant identifier names, and pre-train transformer models for programming language related tasks. Although it does not require any parallel corpora of source code aligned to natural language, methods based on DOBF outperform GraphCodeBERT, CodeBERT and MLM pre-training on multiple downstream tasks, including clone detection, code summarization, natural language code search, and unsupervised code translation. These results show that DOBF leverages the particular structure of source code to add noise to the input sequence in a particularly effective way. Other noise functions or surrogate objectives adapted to source code may improve the performance further. For instance, by training model to find the type of given variables, the signature of a method, or to repair a piece of code which has been corrupted.

Since models pretrained on source code benefit from structured noise, it would be interesting to see whether these findings can be applied to natural languages as well. Although ambiguous, natural languages also have an underlying structure. Leveraging the constituency or dependency parse trees of sentences (as opposed to abstract syntax trees in programming languages) may help designing better pre-training objectives for natural languages.

## Funding Disclosure

The authors of this paper are employed by Facebook France and this work was done using hardware and software provided by Facebook.

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
