Table 3: **Dataset statistics.**

|  | Java | Python |
|---|---|---|
| All - Size | 26 GB | 19 GB |
| All - Nb files | 7.9M | 3.6M |
| Av. nb of tokens / file | 718 | 1245 |
| Av. nb of identifiers / file | 25.9 | 41.8 |

| Input Code | Proposed Function Name | |
|---|---|---|

```java
public static void FUNC_0 (String path){
  try {
    Files.delete(path);
  }
  catch (Exception e) {
    System.err.println("Error deleting file " + path);
  }
}
```

| | |
|---|---|
| deleteFile | 48.3% |
| remove | 16.9% |
| DeleteFile | 13.2% |
| removeFile | 13.1% |
| deleteFileQuietly | 8.4% |

```java
public static void FUNC_0 (String path){
  if (!Files.exists(path)) {
    Files.createDirectories(path);
  }
}
```

| | |
|---|---|
| createDir | 23.5% |
| createDirectory | 20.9% |
| createDirIfNotExists | 20.8% |
| ensureDirectoryExists | 18.5% |
| createDirectoryIfNotExists | 16.3% |

```java
public static List<Pair<String, Double>> FUNC_0 (List<String> list1,
                                                  List<Double> list2)
{
  return IntStream.range(0, Math.min(list1.size(), list2.size()))
              .mapToObj(i -> new Pair<>(list1.get(i), list2.get(i)))
              .collect(Collectors.toList());
}
```

| | |
|---|---|
| zip | 28.6% |
| intersect | 20.0% |
| combine | 17.9% |
| merge | 17.5% |
| intersection | 16.0% |

```java
public static int FUNC_0 (int n){
  int a = 0, b = 1;
  int tmp;
  for (int i = 0; i < n; i ++){
      tmp = a + b;
      a = b;
      b = tmp;
  }
  return a;
}
```

| | |
|---|---|
| fib | 41.5% |
| fibonacci | 36.6% |
| fibon | 9.1% |
| fibo | 8.8% |
| fibonacci_series | 4.0% |

```java
public static float FUNC_0 (List<Float> vec1,
                            List<Float> vec2) {
  float size = vec1.size();
  assert size == vec2.size();
  float result = 0.0f;
  for (int i = 0; i < size; i++) {
    result += vec1.get(i) * vec2.get(i);
  }
  return result;
}
```

| | |
|---|---|
| dotProduct | 40.9% |
| dot | 23.9% |
| dot_product | 16.5% |
| dotproduct | 10.5% |
| inner | 8.3% |

Figure 4: **Examples of name proposal in Java.** DOBF is able to suggest relevant function names for a variety of Java methods and demonstrates its ability to understand the semantics of the code. In the first two examples, the first element in the beam shows that it is able to select relevant names in the context to find a function name: it uses `Files.delete` and `Files.createDirectories` to suggest the tokens `deleteFile` and `createDir`. DOBF finds relevant names for Java methods without copying any part of the other tokens. For example for the third method combining two lists as in the python `zip` function, for the fourth method which computes the n-th element of the Fibonacci series and for the last method which computes the dot product between two vectors.

| Input Code | Proposals for Highlighted Identifiers | |
|---|---|---|
| ```python
def FUNC_0 (name):
  return os.environ[name]
``` | get_env
get_envvar
env
getenv
get_env_variable | 25.3%
19.3%
19.2%
18.5%
17.7% |
| ```python
def FUNC_0 (l):
  return list(set(l))
``` | unique
remove_duplicates
removeDuplicates
uniquify
unique_items | 24.8%
23.8%
18.8%
18.7%
13.8% |
| ```python
def FUNC_0 (path):
  with gzip.open(path, 'rb') as f:
      content = f.read()
  return content
``` | read_gzip_file
read_gzip
ungzip
gzip_content
gzip_read | 22.9%
22.1%
20.8%
18.2%
16.0% |
| ```python
def FUNC_0 (n):
  v = [True for i in range(n + 1)]
  p = 2
  while (p * p <= n):
    if (v[p] == True):
      for i in range(p * 2, n + 1, p):
        v[i] = False
    p += 1
  v[0]= False
  v[1]= False
  return [p for p in range(n+1) if v[p]]
``` | sieve
prime_sieve
sieve_of_eratosthenes
primes
eratosthenes | 36.1%
18.5%
15.5%
15.3%
14.5% |
| ```python
def f(n):
    VAR_0  = [True for i in range(n + 1)]
    p = 2
    while (p * p <= n):
        if ( VAR_0 [p] == True):
            for i in range(p * 2, n + 1, p):
                VAR_0 [i] = False
        p += 1
    VAR_0 [0]= False
    VAR_0 [1]= False
    return [p for p in range(n+1) if VAR_0 [p]]
``` | prime
l
isPrime
a
primes | 30.6%
20.5%
18.0%
16.4%
14.6% |

Figure 5: **Examples of name proposal in Python.** Our model trained with DOBF goes well beyond copying tokens from the context. For instance, in the first example, it understands that this function is used to get environment variables. In the second example, it proposes names related to what this function actually does (removing duplicates in a list) instead of the individual operations it uses (converting to set and then to list). The last two rows show proposals for two different identifiers in a function computing the list of prime numbers below n using the sieve of Eratosthenes. The proposals for the function name are all relevant, and the third one names exactly the algorithm which is used. The variable v is a list of booleans. At the end of the algorithm, v[i] is true if and only if i is prime. The proposed names prime and isPrime are very relevant as they describe what the list contains. Although l and a are not very informative, they indicate that the variable is a list or an array.

| Input Code | Proposed Function Name | |
| --- | --- | --- |
| ```
def FUNC_0 (v1, v2):
  assert len(v1) == len(v2)
  return [a * b for a, b in zip(v1, v2)]
``` | multiply_lists
multiply_list
multiply
multiply_vectors
mul | 28.7%
23.5%
18.1%
14.9%
14.8% |
| ```
def FUNC_0 (v1, v2):
  assert len(v1) == len(v2)
  return sum([a * b for a, b in zip(v1, v2)])
``` | dotproduct
dot_product
dotProduct
dot
multiply_by_addition | 34.8%
19.2%
18.1%
15.6%
12.3% |
| ```
def FUNC_0 (v1, v2):
  assert len(v1) == len(v2)
  return [a ^ b for a, b in zip(v1, v2)]
``` | xor
XOR
vector_xor
xors
xor_lists | 62.9%
12.8%
10.8%
7.4%
6.1% |
| ```
def FUNC_0 (v1, v2):
  assert len(v1) == len(v2)
  return [a ** b for a, b in zip(v1, v2)]
``` | power
list_power
lcm
power_list
powersum | 29.8%
20.9%
19.9%
15.1%
14.3% |
| ```
def FUNC_0 (v1, v2):
  assert len(v1) == len(v2)
  return [a + b for a, b in zip(v1, v2)]
``` | add_lists
add
sum_lists
list_concat
list_add | 27.0%
22.9%
17.9%
17.7%
14.5% |
| ```
def FUNC_0 (v1, v2):
  assert len(v1) == len(v2)
  return [a - b for a, b in zip(v1, v2)]
``` | minus
subtract
difference
subtract_lists
substract | 30.4%
29.8%
14.1%
13.3%
12.4% |

Figure 6: **Examples of function name proposal in Python using DOBF.** DOBF is able to identify the key tokens in each function, to properly infer its purpose, and to suggest appropriate names along with a confidence score. In particular, even though the first two code snippets are very similar in terms of edit distance, they implement very different functions and DOBF is able to name them appropriately.

| BFS Implementation | DFS Implementation | DFS with Erroneous Variable Name |
| --- | --- | --- |
| ```
def FUNC_0 (graph, node):
  visited = [node]
  VAR_0 = [node]
  while VAR_0 :
    s = VAR_0 .pop(0)
    for neighbour in graph[s]:
      if neighbour not in visited:
        visited.add(neighbour)
        VAR_0 .append(neighbour)
  return visited
``` | ```
def FUNC_0 (graph, node):
  visited = [node]
  VAR_0 = [node]
  while VAR_0 :
    s = VAR_0 .pop()
    for neighbour in graph[s]:
      if neighbour not in visited:
        visited.add(neighbour)
        VAR_0 .append(neighbour)
  return visited
``` | ```
def FUNC_0 (graph, node):
  visited = [node]
  queue = [node]
  while queue:
    s = queue.pop()
    for neighbour in graph[s]:
      if neighbour not in visited:
        visited.append(neighbour)
        queue.append(neighbour)
  return visited
``` |
| FUNC_0 bfs \| VAR_0 queue | FUNC_0 dfs \| VAR_0 stack | FUNC_0 bfs |

Figure 7: **Deobfuscation on graph traversal functions.** These three functions perform graph traversals. The only difference between the first and the second function is that the first uses a queue to select the next element (.pop(0)) while the second uses a stack (.pop()). The first function implements a breadth-first search (bfs) in the graph and the second implements a depth-first search (dfs). DOBF is able to find the right function and variable names in each case. In the last function, we replaced the anonymized VAR_0 variable with queue in the implementation of depth-first search. This erroneous information leads DOBF to believe that this function performs breadth-first search. It shows that, just like human programmers, DOBF uses the names of the other variables to understand programs and choose relevant identifier names. When working on code with misleading identifier names, it is often preferable to obfuscate several identifiers.

**Obfuscated Code**

```python
class CLASS_0(nn.Module):

    def __init__(VAR_0, VAR_1, VAR_2, VAR_3):
        super(CLASS_0, VAR_0).__init__()
        VAR_0.VAR_1 = VAR_1
        VAR_0.VAR_2 = VAR_2
        VAR_0.VAR_4 = nn.Linear(VAR_1, (4 * VAR_2), bias=VAR_3)
        VAR_0.VAR_5 = nn.Linear(VAR_2, (4 * VAR_2), bias=VAR_3)
        VAR_0.FUNC_0()

    def FUNC_0(VAR_6):
        VAR_7 = (1.0 / math.sqrt(VAR_6.VAR_8))
        for VAR_9 in VAR_6.VAR_10():
            VAR_9.data.uniform_((- VAR_7), VAR_7)

    def FUNC_1(VAR_11, VAR_12, VAR_13):
        (VAR_14, VAR_15) = VAR_13
        VAR_14 = VAR_14.view(VAR_14.size(1), (- 1))
        VAR_15 = VAR_15.view(VAR_15.size(1), (- 1))
        VAR_12 = VAR_12.view(VAR_12.size(1), (- 1))
        VAR_16 = (VAR_11.VAR_4(VAR_12) + VAR_11.VAR_5(VAR_14))
        VAR_17 = VAR_16[:, :(3 * VAR_11.VAR_8)].sigmoid()
        VAR_18 = VAR_16[:, (3 * VAR_11.VAR_8):].tanh()
        VAR_19 = VAR_17[:, :VAR_11.VAR_8]
        VAR_20 = VAR_17[:, VAR_11.VAR_8:(2 * VAR_11.VAR_8)]
        VAR_21 = VAR_17[:, (- VAR_11.VAR_8):]
        VAR_22 = (th.mul(VAR_15, VAR_20) + th.mul(VAR_19, VAR_18))
        VAR_23 = th.mul(VAR_21, VAR_22.tanh())
        VAR_23 = VAR_23.view(1, VAR_23.size(0), (- 1))
        VAR_22 = VAR_22.view(1, VAR_22.size(0), (- 1))
        return (VAR_23, (VAR_23, VAR_22))
```

**Code Deobfuscated using DOBF**

```python
class LSTM(nn.Module):

    def __init__(self, input_size, hidden_size, bias):
        super(LSTM, self).__init__()
        self.input_size = input_size
        self.hidden_size = hidden_size
        self.h1 = nn.Linear(input_size, (4 * hidden_size), bias=bias)
        self.h2 = nn.Linear(hidden_size, (4 * hidden_size), bias=bias)
        self.init_weights()

    def init_weights(self):
        stdv = (1.0 / math.sqrt(self.hidden_size))
        for m in self.modules():
            m.data.uniform_((- stdv), stdv)

    def forward(self, x, prev_state):
        (prev_h, prev_c) = prev_state
        prev_h = prev_h.view(prev_h.size(1), (- 1))
        prev_c = prev_c.view(prev_c.size(1), (- 1))
        x = x.view(x.size(1), (- 1))
        h = (self.h1(x) + self.h2(prev_h))
        s = h[:, :(3 * self.hidden_size)].sigmoid()
        c = h[:, (3 * self.hidden_size):].tanh()
        r = s[:, :self.hidden_size]
        g = s[:, self.hidden_size:(2 * self.hidden_size)]
        o = s[:, (- self.hidden_size):]
        c = (th.mul(prev_c, g) + th.mul(r, c))
        h = th.mul(o, c.tanh())
        h = h.view(1, h.size(0), (- 1))
        c = c.view(1, c.size(0), (- 1))
        return (h, (h, c))
```

| ID | Ground Truth | DOBF |
|---|---|---|
| CLASS_0 | LSTM | LSTM |
| FUNC_0 | reset_parameters | init_weights |
| FUNC_1 | forward | forward |
| VAR_0 | self | self |
| VAR_1 | input_size | input_size |
| VAR_2 | hidden_size | hidden_size |
| VAR_3 | bias | bias |
| VAR_4 | i2h | h1 |
| VAR_5 | h2h | h2 |
| VAR_6 | self | self |
| VAR_7 | std | stdv |
| VAR_8 | hidden_size | hidden_size |
| VAR_9 | w | m |
| VAR_10 | parameters | modules |
| VAR_11 | self | self |
| VAR_12 | x | x |
| VAR_13 | hidden | prev_state |
| VAR_14 | h | prev_h |
| VAR_15 | c | prev_c |
| VAR_16 | preact | h |
| VAR_17 | gates | s |
| VAR_18 | g_t | c |
| VAR_19 | i_t | r |
| VAR_20 | f_t | g |
| VAR_21 | o_t | o |
| VAR_22 | c_t | c |
| VAR_23 | h_t | h |

Figure 8: **Deobfuscation of an LSTM cell.** DOBF is able to recover several of the original tokens, including the class name (LSTM) and the full signature of the `__init__` method. Even though DOBF does not always recover the original token, it generally proposes very relevant tokens which improves code readability. In particular, for some tokens the accuracy and subtoken scores would be zero but the recovered tokens are still very relevant. For instance, `reset_parameters` (FUNC_0) was renamed to `init_weights`, `std` (VAR_7) was renamed to `stdv`, and `hidden` (VAR_13) was renamed to `prev_state`. In those instances, the original and recovered tokens share no subtoken despite having very similar semantics.

| Input Code | Deobfuscated Identifiers | |
|---|---|---|

```python
def FUNC_0(VAR_0, VAR_1):
    return sum(map(operator.mul, VAR_0, VAR_1))
```
FUNC_0 — dotProduct
VAR_0 — list1
VAR_1 — list2

```python
def FUNC_0(VAR_0):
    VAR_1 = urllib2.urlopen(VAR_0)
    VAR_2 = VAR_1.read()
    return VAR_2
```
FUNC_0 — get_html
VAR_0 — url
VAR_1 — response
VAR_2 — html

```python
def FUNC_0(VAR_0):
    VAR_1 = set(VAR_0)
    return (len(VAR_1) == len(VAR_0))
```
FUNC_0 — all_unique
VAR_0 — iterable
VAR_1 — s

```python
def FUNC_0(VAR_0, VAR_1):
    return list(collections.deque(VAR_0, maxlen=VAR_1))
```
FUNC_0 — tail
VAR_0 — s
VAR_1 — n

```python
def FUNC_0(VAR_0):
    return sum((VAR_1 for VAR_1 in VAR_0 if ((VAR_1 % 2) == 0)))
```
FUNC_0 — even_sum
VAR_0 — nums
VAR_1 — n

Figure 9: **Examples of full deobfuscations of Python functions.** Even when every identifier is obfuscated, DOBF is able to propose relevant names. The proposed function name is informative and relevant in all examples since the first function computes a dot product, the second downloads a HTML page and returns its content, the third evaluates whether the input contains only unique elements, the fourth computes the tail of an iterable, and the fifth computes the sum of the even elements of an iterable.

Table 4: **Results on downstream tasks with the architecture of TransCoder.** This architecture has less layers (6 instead of 12), a higher embedding dimension (1024 instead of 768) and less activation heads (8 instead of 12) resulting in a slightly larger model (143M parameters instead of 126M). It also uses reLU activations instead of geLU. Models pre-trained with MLM and DOBF significantly outperform both CodeBERT and models trained with MLM only. MLM+DOBF outperforms CodeBERT by 7% on natural language code search (NLCS), and MLM by 6% in Java → Python computational accuracy. It also beats CodeBERT on every task except Clone Detection, on which CodeBERT scores much higher than our MLM. GraphCodeBERT only beats our model on python summarization and Python to Java translation by a shallow margin and is below on other tasks. The tasks where MLM provides large improvements over the transformer baseline (first row) are also those where DOBF provides the largest gains (i.e. clone detection, natural language code search, and unsupervised translation).

| | Clone Det (F1 score) | Sum Java (BLEU) | Sum Py (BLEU) | NLCS (MRR) | Py→Ja (CA@1) | | Ja→Py (CA@1) | |
|---|---|---|---|---|---|---|---|---|
| | | | | | k=1 | k=10 | k=1 | k=10 |
| Transformer | 88.14 | 16.58 | 16.43 | 0.025 | 37.6 | 38.9 | 31.8 | 42.1 |
| CodeBERT | 96.50 | 18.25 | 18.22 | 0.315 | - | - | - | - |
| GraphCodeBERT | 96.38 | 18.78 | **18.51** | 0.377 | - | - | - | - |
| MLM | 91.89 | 18.59 | 17.95 | 0.308 | 40.3 | 42.2 | 44.7 | 46.6 |
| DOBF | **96.52** | 18.19 | 17.51 | 0.272 | 38.9 | **45.7** | 44.7 | 46.4 |
| MLM+DOBF | 95.87 | **19.05** | 18.24 | **0.383** | **43.5** | 44.9 | **49.2** | **52.5** |