# OpenReview forum: "DOBF: A Deobfuscation Pre-Training Objective for Programming Languages"
_NeurIPS.cc/2021/Conference — NeurIPS 2021 Poster_

### Official Review · Reviewer_NjSg · 2021-07-12

**Rating:** 7
**Confidence:** 5

**Summary:**

This paper proposes a self-supervised pre-training task: code deobfuscation (recovering erased identifier names). This pre-training task only requires unlabeled natural programs as training data. The authors claim that models pre-trained with code deobfuscation significantly outperform existing approaches on many downstream program understanding tasks, including clone detection, code summarization, NL code search, and unsupervised program translation.

**Main Review:**

This paper proposes a novel pre-training task for natural programs (programs written by humans in real-world projects): code deobfuscation. This idea is not only theoretically reasonable but also empirically effective on both deobfuscation and multiple downstream tasks.

The proposed method is evaluated on multiple well-established public benchmarks, including three subtasks in CodeXGLUE and TransCoder. It is easy to reproduce the results in this paper with the provided source code. Also, the author reruns other baselines to keep the model size similar, making comparisons fair and convincing. Although there are some inconsistent/confusing numbers in the experiment results (see below), the evaluation is sound in general.

This paper includes enough implementation details of the proposed method. It also includes experiments on several variations (different p_obf, with or without MLM) to show how they make every design choice.

However, the usage of DOBF with MLM and DAE puts the effectiveness of DOBF in question.

- MLM: In Table 1, we can see that the best models on two deobfuscation tasks are all initialized with masked language modeling (MLM). In Table 2, "DOBF init MLM" outperforms "DOBF init scratch" on all tasks except Clone Detection, and "DOBF init scratch" is even worse than MLM/CodeBERT on multiple tasks. Therefore, it seems that the combo of MLM and DOBF is what really works, so the author should not overclaim that "DOBF is better than MLM".

- DAE: The authors do not mention DAE at all in the "Model" section, but in Table 2, we find that in all tasks that requires generation (i.e. all tasks except Clone Det), additional DAE training is required to get SOTA performance consistently. However, no enough explanation related to DAE is found in this paper. To make the experiment section more solid, I think "MLM + DAE" and "DAE only" should also be evaluated and reported.

**Inconsistent numbers:** Table 2 reports the results of unsupervised Python/Java translation, where the experimental setting should be the same as the cited TransCoder paper (https://arxiv.org/pdf/2006.03511.pdf). The "MLM" row in Table 2 should correspond to the model in the TransCoder paper. However, the numbers reported here are very different from those reported in the TransCoder paper (the "TransCoder Beam 10 - Top 1" row in Table 2). Moreover, in the TransCoder paper, Java -> Python systems usually have higher computational accuracies; but in this paper, Python -> Java systems has better CA@1. I suggest that the author should explain these differences in detail during the author feedback session.

To summarize, this paper proposed a novel and effective method, has a high-quality experiment section, but still needs to add more explanation in the camera-ready version.

**Time Spent Reviewing:**

10

---

> ### Author Response · Authors · 2021-08-09
> **Response to reviewer NjSg**
>
> Thank you for your feedback and for spending the time to read our paper and the related works thoroughly. We respond to your main concerns below:
>
> **Reviewer comment:**
> MLM: In Table 1, we can see that the best models on two deobfuscation tasks are all initialized with masked language modeling (MLM). In Table 2, "DOBF init MLM" outperforms "DOBF init scratch" on all tasks except Clone Detection, and "DOBF init scratch" is even worse than MLM/CodeBERT on multiple tasks. Therefore, it seems that the combo of MLM and DOBF is what really works, so the author should not overclaim that "DOBF is better than MLM".
>
> **Answer:**
> That is correct. We meant that DOBF is complementary with MLM and that DOBF initialized with MLM outperforms MLM. We clarified it in the updated version of the paper. Thank you for pointing this out.
>
> **Reviewer comment:**
> DAE: The authors do not mention DAE at all in the "Model" section, but in Table 2, we find that in all tasks that requires generation (i.e. all tasks except Clone Det), additional DAE training is required to get SOTA performance consistently. However, no enough explanation related to DAE is found in this paper. To make the experiment section more solid, I think "MLM + DAE" and "DAE only" should also be evaluated and reported.
>
> **Answer:**
> Thank you for pointing that out. We added more information on the DAE in the Model section.
>
> Also, we agree that a comparison with DAE trained from MLM without DOBF is necessary to demonstrate the effectiveness of DOBF.
> We added it as a baseline in our main result table (Table 2) and added references and discussions about this baseline in the text. Here is a link to the new results table: [https://ibb.co/521RNkh](https://ibb.co/521RNkh).
>
> DAE beats MLM on most tasks (all but Java -> Python translation) but still underperforms DOBF + DAE on most tasks. Interestingly, DAE beats every other method for Python -> Java translation, providing small improvements compared to DOBF + DAE: +1.7% and +1.9% CA@1 with beam size 1 and 10 respectively. However, it underperforms every other pre-training method for Java -> Python, with CA@1 8.5% and 9.6% below those of DOBF + DAE with beam size 1 and 10. Overall, adding DOBF during training provides clear gains compared to training with DAE only for several tasks. For code translation, DOBF+DAE is only slightly below DAE for Python -> Java and clearly above for Java -> Python.
>
> **Reviewer comment:**
> Inconsistent numbers: Table 2 reports the results of unsupervised Python/Java translation, where the experimental setting should be the same as the cited TransCoder paper (https://arxiv.org/pdf/2006.03511.pdf). The "MLM" row in Table 2 should correspond to the model in the TransCoder paper. However, the numbers reported here are very different from those reported in the TransCoder paper (the "TransCoder Beam 10 - Top 1" row in Table 2). Moreover, in the TransCoder paper, Java -> Python systems usually have higher computational accuracies; but in this paper, Python -> Java systems has better CA@1. I suggest that the author should explain these differences in detail during the author feedback session.
>
> **Answer:**
> The “MLM” row in Table 2 corresponds to the same pre-training objective as in the TransCoder paper. However, we wanted to compare the DOBF objective to the objectives used in CodeBERT and GraphCodeBERT fairly and decided to use the same architecture and tokenizer as in these papers instead of those from the TransCoder paper. Here are the main differences between our model (identical to the ones of CodeBERT and GraphCodeBERT) and the one from the TransCoder paper:
> - We use RoBERTa’s tokenizer while the TransCoder model uses a custom tokenizer that parses the source code to extract tokens and trains a BPE model on source code.
> - Different architecture. Smaller model: 126M parameters in the encoder versus 143M for TransCoder with more layers (12 instead of 6), a smaller embedding dimension ( 768 instead of 1024) and more activation heads (12 instead of 8). We use geLU activations while they use reLU.
> - We finetune the models for up to 2 days on 32 GPUs while they have no clear time limits.
>
> According to our experiments, this tokenizer and architecture lead to worse results than those from the TransCoder paper for Java -> Python even when keeping the limit on the training time, which is to be expected for a smaller model with a tokenizer designed for NLP. For Python -> Java, the opposite seems to happen. A possible explanation for the low score of TransCoder is that there were a few mistakes in the detokenizer, making the output fail the tests in Java even though the model was outputting the right thing. After fixing it and retraining a model similar with TransCoder’s architecture, we obtained scores significantly higher than those of TransCoder but still lower than from the MLM in our paper (40.3% and 42.2% for Python -> Java). With the version we trained using TransCoder’s architecture and the fixed tokenizer, DOBF initialized with MLM was overperforming TransCoder for every metric (scoring 43.5 and 44.9 for Python -> Java and 49.2 and 52.5 for Java -> Python).
> We agree that the differences between our results and the results of the TransCoder paper can be confusing for the reader and we will clarify that the model we use is different from the one used in the TransCoder paper. We will also add our results with TransCoder’s version of the architecture and tokenizer in the appendix in order to make it clear that models pre-trained with DOBF outperform TransCoder and lead to a new SOTA for unsupervised code Translation. Thank you for your comment.

---

### Official Review · Reviewer_udeg · 2021-07-15

**Rating:** 6
**Confidence:** 5

**Summary:**

This work follows up with earlier efforts to pre-train BERT-like architectures on code, like CuBERT and CodeBERT. Instead of using a language-agnostic masked language model pre-training objective, however, it uses a de-obfuscation objective: masking all occurrences of an identifier (variable, function, class), ask the model to predict it. This code-specific pre-training objective is combined with other recent options (e.g., a denoising autoencoder) as well as traditional mask-language modelling, to produce improved performance on a number of code tasks such as summarization, clone detection, and transpilation.

**Ethical Concerns:**

No concerns.

**Limitations And Societal Impact:**

No concerns.

**Main Review:**

# Overall Evaluation

Originality: This is a relatively incremental idea that seems sensible and appears (at first glance) to improve results.

Quality: There are some concerns on investigating the source of performance improvements and identifying if they're due to the new contribution (DOBF), more samples trained on, or ancillary additions (e.g., DAE). A bit of experimental refinement should resolve the issues. Some concerns also with information leak across pre-training and fine-tuning datasets, that could be clarified during rebuttal.

Clarity: The submission is clearly written and related work is well presented and contrasted.

Significance: Although the new nugget here is relatively small, assuming the evaluation is strengthened, it could be useful in lots of future work in the space of ML for code.



# Main Comments

This work suggests a sensible improvement over the prior art on unsupervised pre-training on code. You make a good case with traditional MLM has a mix of very easy and not-so-easy instances in code contexts. Of course, the same argument applies to natural language as well (eg., predicting articles, pronouns, expressions is much easier than the typical cloze predictions), so this is a sensible thing to try, albeit not particular surprising.

This work involved a lot of engineering (getting the DOBF implementation to play with the CodeXGLUE tasks and TransCoder), and this is quite appreciated. But the novel contribution here doesn't contain much in terms of modelling. All are known building blocks put together to implement and evaluate a simple idea.

I had some concerns about the evaluation that might be possible to address during rebuttals.

1. MLM still helps DOBF, so the "easy" cases still seem to provide value. Given that MLM still seems to improve with even more training, what would happen if you produced twice as many MLM examples (say using additional random seeds or going beyond the 15% MLM tokens or 20 MLM tokens cap from traditional BERT) instead of MLM + DOBF? Is it truly the case that using more MLM examples and instances would never catch up?

2. DAE seems to be an orthogonal addition to the picture (and not a contribution here, other than just trying it out on code), that could compose with MLM just as well as it does with DOBF. Would MLM+DAE provide a similar boost to MLM? More?

2. It's not clear what the sample cost of each row on Table 2 is (this is related to the above). Is DOBF + DAE pre-training on more samples than DOBF alone or MLM alone? Fixing the number of samples across all combinations seems like a more rational way to compare the approaches on their merits, rather than making an uncontrolled comparison.

1. The differences about some of the numbers in Table 2 are small. Confidence intervals are very important to draw sound conclusions here. Please add confidence intervals.

1. I'm not entirely clear on Table 1's metrics. It seems your metrics are per deobfuscation instance, treating samples with say 10 de-obfuscations the same as samples with 1. Is that true? Are you measuring any performance with respect to the per-sample F1 score of token de-obfuscations? This is a weak concern, since this is a pre-training task, and ultimately it's the fine-tuning tasks we care about. But for a de-obfuscation task as a first-class outcome, we'd want to know how well a single sample is deobfuscated, not just the deobfuscated variables in the test dataset. Please clarify your metrics.

1. In your experimental details, you mention that you deduplicate your pre-training dataset, but you also mention "duplicate files". Does that mean you're only removing identical files? Or do you use the set/multi-set similarity scores of Allamanis to remove even near-duplicate files? Please specify what you do exactly.

1. Also, what about your fine-tuning datasets? Do you ensure that none of your fine-tuning samples (in either fine-tuning split) appear exactly or approximately in your pre-training dataset? In other words, do you do what Kanade et al. did for cubert to remove all fine-tuning files from the pre-training files? Otherwise, you may easily end up with severe label leaking problems.

1. You mention in the fine-tuning details that you train models with the same architectures, batch sizes, optimizers, etc. I'd like to understand if that means you performed hyperparameter searches that include all the same sizes (but still chose the best for each)? Or did you just choose the same hyperparameters and just search across learning rates? The issue here is that fairness doesn't mean that all baselines have the exact same architecture; it is, instead, that all baselines are *given the chance* to be best with the same architecture (the same number of trainable parameters, hidden sizes, etc) even if for the particular dataset, the best choice might not be the same choice for each.



# Smaller Comments

* Line 49--53: This claim seems false, given lines 304 and 305, where you point out that you use MLM to initialize DOBF and found that initializing DOBF from scratch was inferior. It seems you need both for good performance.



**Time Spent Reviewing:**

4

---

> ### Author Response · Authors · 2021-08-09
> **Response to reviewer udeg (1/2)**
>
> Thank you for your thorough feedback. We respond to your main concerns below:
>
> 1. MLM still helps DOBF, so the "easy" cases still seem to provide value. Given that MLM still seems to improve with even more training, what would happen if you produced twice as many MLM examples (say using additional random seeds or going beyond the 15% MLM tokens or 20 MLM tokens cap from traditional BERT) instead of MLM + DOBF? Is it truly the case that using more MLM examples and instances would never catch up?
>
> **Answer:**
> Increasing the number of masked tokens would make the objective more difficult but would not provide the same benefits as DOBF. Even though MLM can mask all the occurences of a variable, the probability that it does is low for variables that appear in several places in the file and the model can learn to retrieve the masked tokens without generating a good representation of the semantics of the code. For instance, in the example with depth-first search in our paper, the model learns to pay attention to the variables used in the function when deciding how to call the queue variable in `[MASK] = [root]` as MLM would be unlikely to mask all its occurrences. If we were to increase the masking probability to make it likely, we would also be likely to mask the semantics necessary to guess the right variable name (if the 0 in `.pop(0)` is masked, there is no way to know that it was not a stack with `.pop(-1)`). With DOBF, the model needs to pay attention to the semantics of the code (here the `.pop(0)` call), which are kept intact.
>
> 2. DAE seems to be an orthogonal addition to the picture (and not a contribution here, other than just trying it out on code), that could compose with MLM just as well as it does with DOBF. Would MLM+DAE provide a similar boost to MLM? More?
>
> The comparison with DAE trained from MLM without DOBF is indeed necessary to demonstrate the effectiveness of DOBF. Thank you for pointing this out. We added it as a baseline in our main result table (Table 2) and added references and discussions about this baseline in the text. Here is a link to the new results table: [https://ibb.co/521RNkh](https://ibb.co/521RNkh).
>
> DAE beats MLM on most tasks (all but Java -> Python translation) but still underperforms DOBF + DAE on most tasks. Interestingly, DAE beats every other method for Python -> Java translation, providing small improvements compared to DOBF + DAE: +1.7% and +1.9% CA@1 with beam size 1 and 10 respectively. However, it underperforms every other pre-training method for Java -> Python, with CA@1 8.5% and 9.6% below those of DOBF + DAE with beam size 1 and 10. Overall, adding DOBF during training provides clear gains compared to training with DAE only for several tasks. For code translation, DOBF+DAE is only slightly below DAE for Python -> Java and clearly above for Java -> Python.
>
> 3. It's not clear what the sample cost of each row on Table 2 is (this is related to the above). Is DOBF + DAE pre-training on more samples than DOBF alone or MLM alone? Fixing the number of samples across all combinations seems like a more rational way to compare the approaches on their merits, rather than making an uncontrolled comparison.
>
> **Answer:**
> We trained all our models until convergence (no improvements on the validation set for 10 consecutive epochs), which always happened in less than 8 days (our time limit) on 32 GPUs for our model size. DOBF + DAE was treated equally to DOBF and DAE alone (with this protocol, DAE alone could potentially take longer to train than DOBF + DAE). These models are initialized using our MLM model so they were trained longer and with more samples than our MLM baseline. However, since our MLM baseline had already converged, training it longer should not result in a better performance.
>
> 4. The differences about some of the numbers in Table 2 are small. Confidence intervals are very important to draw sound conclusions here. Please add confidence intervals.
>
> **Answer:**
> We agree that confidence intervals in Table 2 would be helpful. Unfortunately, each model took days to train on dozens of GPUs, so computing confidence intervals with bootstraps would be extremely expensive. We believe that this is the reason why there are no confidence intervals in the papers implementing our baselines either.
>
> We agree that some of the differences between the numbers in Table 2 are small, however we believe our method is clearly above the CodeBERT and GraphCodeBERT baselines for many tasks, such as Java code summarization, NL code search and code translation. Moreover, the performances of our models are quite consistent across tasks, which helps to evaluate their value as pre-trained models.
>
> 5. I'm not entirely clear on Table 1's metrics. It seems your metrics are per deobfuscation instance, treating samples with say 10 de-obfuscations the same as samples with 1. Is that true? Are you measuring any performance with respect to the per-sample F1 score of token de-obfuscations? This is a weak concern, since this is a pre-training task, and ultimately it's the fine-tuning tasks we care about. But for a de-obfuscation task as a first-class outcome, we'd want to know how well a single sample is deobfuscated, not just the deobfuscated variables in the test dataset. Please clarify your metrics.
>
> **Answer:**
> These metrics measure the average subtoken F1 score and exact-match accuracy for files in the test set. For instance, with DOBF + MLM trained with p_obf = 0.5 (the model we selected), for a random file in the test set, in expectation the model will retrieve 45.7% of the original tokens exactly and will retrieve the sub-tokens with a f1 score of 58.0. Therefore, this metric gives you some information about how well a random sample (file) is deobfuscated in expectation. We agree that the description of our metric is ambiguous. We updated the paper to explain that our metrics are given as averages per file.
> We are not entirely sure that our metric is the metric you are asking for so please tell us if you still have any concern about the metrics.
>
> 6. In your experimental details, you mention that you deduplicate your pre-training dataset, but you also mention "duplicate files". Does that mean you're only removing identical files? Or do you use the set/multi-set similarity scores of Allamanis to remove even near-duplicate files? Please specify what you do exactly.
>
> **Answer:**
> We deduplicate on the standardized and obfuscated inputs. We standardize the whitespaces, newlines and indentation and the obfuscation standardizes the identifier names. Compared to Allamanis, our duplicate removal scheme focuses on the structure and not on the identifiers and literals. We did not want to use their deduplication scheme for the following reasons:
> - We want our model to be robust to noise and to be able to assign similar embeddings to code snippets that are considered to be near duplicates and removed by Allamanis (codes with similar identifier names but different structures). We believe these files are beneficial for pre-training our model for downstream tasks.
> - Real users are likely to want to deobfuscate near-duplicate files if they are common in real code. Therefore it is not clear that measuring the accuracy and f1 scores on a dataset without near-duplicates would be a better representation of the model’s value for real users. We consider that studying which metric best correlates with human appreciation for deobfuscation and name proposal is outside of the scope of this paper since we mostly focus on DOBF as a pre-training objective.
>
> 7. Also, what about your fine-tuning datasets? Do you ensure that none of your fine-tuning samples (in either fine-tuning split) appear exactly or approximately in your pre-training dataset? In other words, do you do what Kanade et al. did for cubert to remove all fine-tuning files from the pre-training files? Otherwise, you may easily end up with severe label leaking problems.
>
> **Answer:**
> Ideally, we would want to remove all the files containing functions similar to those in any of the test sets for the downstream tasks from our pre-training dataset. However, this seems very difficult to do in practice. For instance, in the case of TransCoder, for an input function in (say) Java, we would need to remove from our GitHub dataset all Python functions that implement the same thing as this Java function, and not just the translation in our test set. Moreover, removing files is probably not enough, as some files may have been split and we would need to consider their content at function level.
>
> To the best of our knowledge, our baselines (CodeBERT and GraphCodeBERT) do not remove all fine-tuning files from the pre-training files, and we did not either.
>
> The pre-training objectives are very different from the downstream task, and we do not pre-train on the labels of the downstream tasks, so we believe that the risk of label leaking is limited and that it would not impact the results anyway. As some of the fine-tuning tasks involve matching the code with the docstring or comments (e.g. code summarization or code search), we removed all docstrings and comments from the pre-training set to mitigate the risks.
>
> Besides, the main goal of the paper is to compare different pretraining objectives, and all our models (MLM, DAE, DOBF, DOBF + DAE) are pretrained on the same dataset, so it should not impact the fairness of the comparison between our experiments and our baselines.

---

> > ### Author Response · Authors · 2021-08-09
> > **Response to reviewer udeg (2/2)**
> >
> > 8. You mention in the fine-tuning details that you train models with the same architectures, batch sizes, optimizers, etc. I'd like to understand if that means you performed hyperparameter searches that include all the same sizes (but still chose the best for each)? Or did you just choose the same hyperparameters and just search across learning rates? The issue here is that fairness doesn't mean that all baselines have the exact same architecture; it is, instead, that all baselines are given the chance to be best with the same architecture (the same number of trainable parameters, hidden sizes, etc) even if for the particular dataset, the best choice might not be the same choice for each.
> >
> > **Answer:**
> > Early experiments we ran when fine-tuning on downstream tasks suggested that the learning rate was by far the most sensitive parameter. As a result, for each model, we performed a hyper-parameter search only on the learning rate and selected the best using the validation score. We chose the other hyperparameters for each downstream task independently from our models (i.e. as the default parameters given in the examples on CodeXGlue’s repository or TransCoder’s default parameters). We believe we have given the same chance to each model to perform best with the same architecture.
> >
> > **Reviewer comment:** Line 49--53: This claim seems false, given lines 304 and 305, where you point out that you use MLM to initialize DOBF and found that initializing DOBF from scratch was inferior. It seems you need both for good performance.
> >
> > **Answer:**
> > Thank you for pointing this out. We updated the text to make it clearer that MLM is useful and that MLM and DOBF are complementary.

---

> > > ### Comment · Reviewer_udeg · 2021-08-29
> > > **Response response 2/2**
> > >
> > > Thanks for these answers.

---

> > ### Comment · Reviewer_udeg · 2021-08-29
> > **Response response 1/2**
> >
> > ```
> > 1. ... Increasing the number of masked tokens would make the objective more difficult but would not provide the same benefits as DOBF. Even though MLM can mask all the occurences of a variable, the probability that it does is low for variables that appear in several places in the file and the model can learn to retrieve the masked tokens without generating a good representation of the semantics of the code. ...
> > ```
> > Ah, perhaps you misunderstood me. I wasn't arguing that if you increase MLM samples or masked tokens per sample, you might "simulate" DOBF, and therefore subsume it. I agree with you that just through chance, having the MLM sampler produce DOBF samples sounds unlikely.
> >
> > What I am asking is if the MLM objective along, if given enough (more) distinct samples, can perhaps produce as good a pre-trained model, without adding the special DOBF objective.
> >
> > Just to address your response further: partially masking a few occurrences of a variable and having the model recover them as such isn't a trivial objective, and I'd argue it's a harder objective than deobfuscation. That's a hybrid variable misuse/variable naming task (mask a variable and choose which of the existing variable names, or a new one, it corresponds to). It may not be a traditional deobfuscation task, but traditional deobfuscation has the easier job of dealing with code that must still run (e.g., obfuscated binaries, apps, etc.). If you are not bound by that constraint (and you are not in the pre-training scenario), you can have far more challenging partial obfuscations (and, of course, corresponding pre-training examples). My point is that having samples of a partially obfuscated variable is not inherently easier for the model, or less valuable in learning structure. Therefore I question your claim that having more MLM samples is by construction weaker, just because it is unlikely to simulate deobfuscation samples.
> >
> > So, to get back to my question above: you currently train MLM with some pre-training dataset of size A (to convergence, presumably, and then further train the resulting model with a DOBF dataset of size B (again, to convergence); if you were instead to generate a different MLM' dataset of size A+B and let it converge, would the model fine-tuned on MLM' be worse than the model fine-tuned on MLM+DOBF?
> >
> > ```
> > 2. ...
> > ```
> > Thanks for the extra experiments. They help!
> >
> > ```
> > 3. ... We trained all our models until convergence (no improvements on the validation set for 10 consecutive epochs), which always happened in less than 8 days (our time limit) on 32 GPUs for our model size. DOBF + DAE was treated equally to DOBF and DAE alone (with this protocol, DAE alone could potentially take longer to train than DOBF + DAE).
> > ```
> > It's good that the training protocol is the same, but it doesn't answer my question. I was asking about the training dataset size. More data leads to better models (as evidenced by the improvements in pre-trained model quality with ever larger corpora). So what I'm asking is this: is the DOBF+DAE model seeing more distinct samples than the number of distinct samples seen by the DOBF model alone, or the DAE model alone? Just in case it helps with clarity: for MLM, given 1M source files (and ignoring any splitting of long files), you can generate 10M MLM training samples if you draw 10 MLM samples from each file, or 100M MLM training samples if you draw 100 MLM samples per file. Given that you train for multiple epochs, every distinct sample is seen many times by each model, but what I want to know is how many unique samples does each model see, and is that number the same for all models. If not, the models that see more distinct samples are given an advantage.
> >
> > ```
> > 4. ... so computing confidence intervals with bootstraps would be extremely expensive...
> > ```
> > Agreed, pre-training 10 times would be very expensive. However, fine-tuning 10 times might not be. Or even computing test results on 10 subsampled test corpora. The dataset sizes seems to indicate that at least for test datasets there's room to do some bootstrap CI computation.
> >
> > ```
> > 6. ...
> > ```
> > Agreed, deduplication is a very weak concern for the pre-training corpus. Thanks for the explanation.
> >
> > ```
> > 7.  ... Ideally, we would want to remove all the files containing functions similar to those in any of the test sets for the downstream tasks from our pre-training dataset. However, this seems very difficult to do in practice. For instance, in the case of TransCoder, for an input function in (say) Java, we would need to remove from our GitHub dataset all Python functions that implement the same thing as this Java function, and not just the translation in our test set. ...
> > ```
> > The standard in the literature is avoiding having a pre-training example leak the label of a fine-tuning example. Removing from the pre-training datasets functions that are lexically similar to a fine-tuning example seems essential in that case. Otherwise, you have no idea if the pre-trained model memorized the necessary information to answer the fine-tuning task. If you're fine-tuning on variable naming, and some pre-training example contains the summary and enough of the context, then that's a problem with your methodology.
> >
> > `all Python functions that implement the same thing` would perhaps be overkill in this case. Allamanis is only talking about things that look similar (in terms of identifiers) to fine-tuning samples. So in the transpilation case, that would be removing all pre-training samples that have a high Jaccard similarity to either the Python function or the Java function in the sample. That doesn't seem unreasonable.
> >
> > ```
> > To the best of our knowledge, our baselines (CodeBERT and GraphCodeBERT) do not remove all fine-tuning files from the pre-training files, and we did not either.
> > ```
> > There's definitely the example of CuBERT that did remove all fine-tuning files (and lexically similar files) from their pre-training corpus, so this is certainly not unprecedented, and seems like prudent practice.
> >
> > ```
> > As some of the fine-tuning tasks involve matching the code with the docstring or comments (e.g. code summarization or code search), we removed all docstrings and comments from the pre-training set to mitigate the risks.
> > ```
> > That is definitely a good choice. Thanks for clarifying that. Perhaps what might make sense is to point out in the paper that for the specific four fine-tuning tasks you selected, you removed the relevant parts of the pre-training samples to avoid label leaking.
> >
> > ```
> > Besides, the main goal of the paper is to compare different pretraining objectives, and all our models (MLM, DAE, DOBF, DOBF + DAE) are pretrained on the same dataset, so it should not impact the fairness of the comparison between our experiments and our baselines.
> > ```
> > Modulo my question above about dataset sizes, if the different pre-trained model are not seeing the same number of samples, they might get unfair advantages, if the fine-tuning samples are not removed from the corpus used to pre-train. Even if the label isn't leaked, having some function seen de-obfuscated, DAE'ed, MLM'ed a bunch of times during pre-training might help the resulting model do a better/worse job when fine-tuning on a sample with that function, than if it had never seen this function before. And if DOBF+DAE has seen more samples of that function than MLM, or DOBF alone, that's an unfair advantage. So the prudent (and conservative) thing to do is to just remove all those similar samples, so such questions cannot arise.

---

> > > ### Author Response · Authors · 2021-09-02
> > > **Second response to reviewer udeg**
> > >
> > > >Ah, perhaps you misunderstood me. I wasn't arguing that if you increase MLM samples or masked tokens per sample, you might "simulate" DOBF, and therefore subsume it. I agree with you that just through chance, having the MLM sampler produce DOBF samples sounds unlikely.
> > > >What I am asking is if the MLM objective along, if given enough (more) distinct samples, can perhaps produce as good a pre-trained model, without adding the special DOBF objective.
> > > >So, to get back to my question above: you currently train MLM with some pre-training dataset of size A (to convergence, presumably, and then further train the resulting model with a DOBF dataset of size B (again, to convergence); if you were instead to generate a different MLM' dataset of size A+B and let it converge, would the model fine-tuned on MLM' be worse than the model fine-tuned on MLM+DOBF? ...for MLM, given 1M source files (and ignoring any splitting of long files), you can generate 10M MLM training samples if you draw 10 MLM samples from each file, or 100M MLM training samples if you draw 100 MLM samples per file. Given that you train for multiple epochs, every distinct sample is seen many times by each model, but what I want to know is how many unique samples does each model see, and is that number the same for all models. If not, the models that see more distinct samples are given an advantage.
> > >
> > > Thank you for clarifying your question, we indeed misunderstood you at first.
> > > What we mean by “training dataset” in the paper and in our other answers is the set of github files we train on and our “dataset size” is the number of distinct github files we train on. This training dataset is the same for MLM and DOBF. Of course, the actual inputs and outputs of the model are generated randomly and will be different for MLM and DOBF. However, we don’t generate a fixed number of training examples. Instead, we sample them randomly on the fly for each batch. For instance, for MLM, when an input sample comes in, we generate the mask randomly and train the model to retrieve the masked tokens. When this example appears again during the second epoch, the mask will be sampled again and will probably be different from the mask sampled during the first epoch. Since we train the MLM model until convergence and on (mostly) different inputs and outputs since the masks are sampled randomly at each epoch, we could not improve it significantly by training it longer or sampling new masks and we believe our setting provides a fair comparison between our methods (i.e. for each configuration we optimized and trained to obtain the best performance).
> > >
> > > >Just to address your response further: partially masking a few occurrences of a variable and having the model recover them as such isn't a trivial objective, and I'd argue it's a harder objective than deobfuscation ... My point is that having samples of a partially obfuscated variable is not inherently easier for the model, or less valuable in learning structure. Therefore I question your claim that having more MLM samples is by construction weaker, just because it is unlikely to simulate deobfuscation samples.
> > >
> > > We agree with you that the MLM objective is not always trivial and can be useful to train models for code and, as you noticed, you can find some examples where finding the right variable to use is a difficult and interesting task. Our experiments also show that pre-training models with MLM is beneficial. We still noticed that most of the exemples generated with MLM are relatively easy. It is noticeable qualitatively when looking at a few of those examples and also quantitatively when looking at the perplexity of MLM trained on code vs on natural languages: we quickly reach a perplexity of 1.4 on code while RoBERTa converges around 3.6 on natural languages. Also, the average crossentropy loss per retrieved token with the DOBF objective at convergence is about 3 times that of MLM in our experiments, even though some of the tokens DOBF produces to retrieve the dictionary are very easy to find (e.g. `VAR_i` tokens, separator tokens `|` between two dictionary entries), showing that the DOBF task is more difficult in average.
> > >
> > >
> > > >The standard in the literature is avoiding having a pre-training example leak the label of a fine-tuning example. Removing from the pre-training datasets functions that are lexically similar to a fine-tuning example seems essential in that case. Otherwise, you have no idea if the pre-trained model memorized the necessary information to answer the fine-tuning task. If you're fine-tuning on variable naming, and some pre-training example contains the summary and enough of the context, then that's a problem with your methodology.
> > >
> > > >all Python functions that implement the same thing would perhaps be overkill in this case. Allamanis is only talking about things that look similar (in terms of identifiers) to fine-tuning samples. So in the transpilation case, that would be removing all pre-training samples that have a high Jaccard similarity to either the Python function or the Java function in the sample. That doesn't seem unreasonable.
> > > >There's definitely the example of CuBERT that did remove all fine-tuning files (and lexically similar files) from their pre-training corpus, so this is certainly not unprecedented, and seems like prudent practice.
> > > >Modulo my question above about dataset sizes, if the different pre-trained model are not seeing the same number of samples, they might get unfair advantages, if the fine-tuning samples are not removed from the corpus used to pre-train. Even if the label isn't leaked, having some function seen de-obfuscated, DAE'ed, MLM'ed a bunch of times during pre-training might help the resulting model do a better/worse job when fine-tuning on a sample with that function, than if it had never seen this function before. And if DOBF+DAE has seen more samples of that function than MLM, or DOBF alone, that's an unfair advantage. So the prudent (and conservative) thing to do is to just remove all those similar samples, so such questions cannot arise.
> > >
> > > We agree that it is important to avoid having test labels leaking in the training set and that the efforts of the authors of CuBERT to avoid this issue are commendable. However, their methodology is rather exceptional than standard in the model pre-training litterature and they may have been motivated by their particular fine-tuning tasks (finding whether variable calls or operands were swapped is easier if you’ve seen the real function). Influential pre-training models such as BERT, RoBERTa and our baselines CodeBERT and GraphCodeBERT do not remove the samples which are the same or similar to samples in their fine-tuning tasks from their train set. We will explain why we strongly believe that doing such a thing is not necessary to show the value of either their models or ours.
> > > While we agree that there would be an issue in our methodology if we were fine-tuning on a variable naming task, it is not the case here and we do not leak the labels for any of our tasks. Please find the detailed explanations for each task below:
> > > 1. Code summarization (Python and Java). The goal for this task is to generate a docstring given the corresponding code snippet. We removed all the comments and docstrings from our pre-training dataset, so the model cannot possibly learn to link the function to the docstring for any of the samples in the test set of the fine-tuning task and does not get any information or label which is not in the input for the fine-tuning task.
> > > 2. Natural language code search. The goal is to find the code snippet corresponding to a natural language query (docstring). For the same reason as for the code summarization tasks, the labels cannot be leaked for this task.
> > > 3. Clone detection. The goal of this task is to classify whether two code snippets have the same semantics. We don’t give such information to the model when pre-training and do not give any information that is not available in the test set samples when pre-training.
> > > 4. Code translation. Unsupervised method to translate a function to a function with the same semantics written in a different programming language. Some correct translations of our input functions may be in our pre-training set and that may help the model to generate correct translations. However, the model doesn’t see any translation examples and there is no reason to believe that the metric we compute would not reflect the value of the model for translating other functions (even a model learning only to retrieve the right python function from a database for a given input java function would be valuable in that case). In any case, removing every possible translation of every input function in the test dataset would be a very difficult task.
> > >
> > > Moreover, we trained our MLM, DOBF, Denoising  and DOBF+Denoising models until convergence and the masks (for MLM and denoising), obfuscated variables (for DOBF) and noise (for denoising) are sampled randomly on the fly for each batch so that we don’t arbitrarily limit the number of samples any model can see. Even if we assume that there is a large advantage to be gained from examples similar to those of the test set of the fine-tuning tasks in the train set, all these models could benefit from it and the comparison between these pre-trained models is still fair and valid.

---

### Official Review · Reviewer_unSs · 2021-07-15

**Rating:** 8
**Confidence:** 5

**Summary:**

This paper produces a new pretraining objective for models that provide program embeddings via the use of a name deobfuscation objective. This objective has advantages over the standard MLM objective as it excludes obvious masking possibilities and also does not allow the model to copy the names of one token from part of the document to another part.

The authors construct this objective and demonstrate that their model can learn this task fairly effectively. Interestingly, initializing their objective with a model pretrained on MLM seems to aid the performance of their model. They also demonstrate improvement over some baselines on some standard program processing objectives, and improvement over all baselines using a combination of their objective and a separate program recovery objective.


**Limitations And Societal Impact:**

I agree with the authors that research on deobfuscation does more good than harm, but disagree with their reasoning about this work in particular. In general, the kinds of obfuscation performed by socially desirable/legal and socially undesirable/criminal actors are fairly similar and thus helping solve some but not all obfuscation tasks does not harm one group more than the other. I think in general the reason this kind of work is socially desirable is because the issue of reverse engineering of software is targetable legally whereas deobfuscation for virus detection, etc., is a primarily technical problem.


**Main Review:**

Originality: this paper’s deobfuscation objective seems interesting and novel.

Quality: The paper’s analysis of this new task and the techniques used to solve it directly seem sound, as does the inclusion of MLM training and how it improves the results.

One problematic element of this paper is the inclusion of DAE as an additional objective on top of DOBF. As it is an element added in to the work on top of MLM, it should probably also be included as a baseline as it is clearly providing a great degree of support to the work. In table 2, if DAE were to be removed, the DOBF models would not provide the best result in two of the six settings (Code Sum Python and Python → Java). Thus I think it is important to include a DAE baseline so we can be sure it is DOBF and not DAE providing the benefits.

Clarity: The paper is generally clearly written and goes over the topics in sufficient detail to be understood.

One minor criticism is that the word “deobfuscation” as used throughout this paper is substantially narrower than what people might picture when they hear the word being used. An explicit sentence about this in the introduction perhaps in the paragraph on obfuscation would help a reader understand what is going on in the paper

Minor clarity complaints:
280: if possible please move the table closer


========================

Edit (following reply to this review): the DAE-onnly baseline was helpful in improving the robustness of the paper. My score has been adjusted correspondingly.

**Time Spent Reviewing:**

1.25

---

> ### Author Response · Authors · 2021-08-09
> **Response to reviewer unSs**
>
> Thank you for your feedback and your excellent suggestions. We respond to your main concerns below:
>
> **Reviewer comment:** "I think it is important to include a DAE baseline so we can be sure it is DOBF and not DAE providing the benefits."
>
> **Answer:**
> Thank you for your remark.
> The comparison with DAE trained from MLM without DOBF is indeed necessary to demonstrate the effectiveness of DOBF. Thank you for pointing this out. We added it as a baseline in our main result table (Table 2) and added references and discussions about this baseline in the text. Here is a link to the new results table:  [https://ibb.co/521RNkh](https://ibb.co/521RNkh).
>
> DAE beats MLM on most tasks (all but Java -> Python translation) but still underperforms DOBF + DAE on most tasks. Interestingly, DAE beats every other method for Python -> Java translation, providing small improvements compared to DOBF + DAE: +1.7% and +1.9% CA@1 with beam size 1 and 10 respectively. However, it underperforms every other pre-training method for Java -> Python, with CA@1 8.5% and 9.6% below those of DOBF + DAE with beam size 1 and 10. Overall, adding DOBF during training provides clear gains compared to training with DAE only for several tasks. For code translation, DOBF+DAE is only slightly below DAE for Python -> Java and clearly above for Java -> Python.
>
>
> **Reviewer comment:** "One minor criticism is that the word “deobfuscation” as used throughout this paper is substantially narrower than what people might picture when they hear the word being used. An explicit sentence about this in the introduction perhaps in the paragraph on obfuscation would help a reader understand what is going on in the paper"
>
> **Answer:** We agree that this term may be confusing without any extra clarifications. As suggested, we modified the introduction to make our definition of “deobfuscation” clearer.
>
> **Reviewer comment:**
> "Minor clarity complaints: 280: if possible please move the table closer"
>
> **Answer:**
> Thank you. We made sure to use only single spaces between words and a shorter space (~ in latex) between Table and 1.
>
> **Reviewer comment:**
> "I agree with the authors that research on deobfuscation does more good than harm, but disagree with their reasoning about this work in particular. In general, the kinds of obfuscation performed by socially desirable/legal and socially undesirable/criminal actors are fairly similar and thus helping solve some but not all obfuscation tasks does not harm one group more than the other. I think in general the reason this kind of work is socially desirable is because the issue of reverse engineering of software is targetable legally whereas deobfuscation for virus detection, etc., is a primarily technical problem."
>
> **Answer:**
> Thank you, this is a very interesting argument and we modified our paragraph on the societal impact of our work taking your comment into account.

---

> > ### Comment · Reviewer_unSs · 2021-08-17
> > **Inclusion of DAE baseline**
> >
> > Thank you for the inclusion of the DAE baseline. It is interesting that DAE and DOBF + DAE are so different for the two directions of Python <-> Java translation but the overall results are clearly positive for DOBF + DAE.

---

### Official Review · Reviewer_K81R · 2021-07-16

**Rating:** 5
**Confidence:** 4

**Summary:**

This paper proposes a pre-training method for programming languages. The main idea is pre-training a seq2seq model to convert obfuscated functions back to their original forms. The method demonstrates performance improvement on multiple downstream tasks, e.g., code translation and code search.

**Limitations And Societal Impact:**

The authors have addressed the limitations and potential negative societal impact (e.g., theft of proprietary code) of their work.

**Main Review:**

The proposed pre-training objective, named DOBF,  is based on the classical masked language model (MLM). The main differences with MLM lie in that (1) DOBF only masks variable, method, and class identifiers rather than syntax-related tokens (e.g., parenthesis and semi-column), and  (2) all occurrences of a masked variable will be replaced by the same special token.

My main concern is that the motivation behind DOBF is not well justified. The authors describe the limitations of MLM for programming languages in Lines 50-56, where some critical claims are somewhat inaccurate.

* They claim that syntax-related predictions "provide little training signal to the model," but I guess these predictions do help to capture **structure** semantics. That may also be the main reason for the poor performance of DOBF without MLM initialized. Besides, the authors state that "DOBF leverages the structural aspect of programming languages" in the abstract, but I can not see how DOBF leverages the **structural aspect**. Anyway, variables are more about ambiguous information instead of structure information.

* DOBF masks all occurrences of a variable with the same special token since the authors believe "it will be easy for the model to simply copy its name from one of the other occurrences." From my point of view, learning to copy which identifier to the masked position is a non-trivial task, which will help capture contextual information. However, masking with the same token will tell the model that the two variable is the same. I believe both masking strategies have their merits and they are somewhat complementary, but there is no in-depth evaluation to compare them. I see the overall performance improvement, but I am wondering whether this strategy works.

I am very interested in the experimental results of deobfuscation. According to a previous code summarization work [1], it is non-trivial to generate meaningful code summaries when removing all variable identifiers in a function.   Though generating code summaries is different from recovering masked variables,  it is surprising that DOBF_1 recovers 45.6% of the initial identiﬁer names, and the recovered code in Figure 2 looks perfect. How do you split the training set and test set for this evaluation?  Maybe there is something that I misunderstand, but I guess the surprising performance may attribute to the memorization of duplicated code rather than the generalization abilities.

Regarding the experiments of the downstream tasks, the overall performance improvements over strong baselines (e.g., GraphCodeBERT) look like mainly coming from DAE, which is not the technical contribution of this paper. Without DAE, DOBF demonstrates no obvious superiority.

Besides, there has been a pre-training method to predict a function name [2]. The idea is not quite new given the current technical trend, though it is a more general method targeting more downstream tasks of programming languages. [2] should be an important missing reference.

Overall, this paper is an incremental study, and the evaluation is weak, especially for a high-profile conference. Investigating further and more fine-grained how and why DOBF improve over MLM could improve this work to a more solid one.

-----After the authors' rebuttal-----

The response generally makes sense. This work's technical novelty and overall quality look more fit for a short paper. I will keep my score.


[1] A neural model for generating natural language summaries of program subroutines, ICSE 2019

[2] Exploiting Method Names to Improve Code Summarization: A Deliberation Multi-Task Learning Approach, ICPC 2021


**Time Spent Reviewing:**

4

---

> ### Author Response · Authors · 2021-08-09
> **Response to reviewer K81R**
>
> Thank you for your feedback and suggestions. We respond to your main concerns below:
>
> **Reviewer comment:**
> They claim that syntax-related predictions "provide little training signal to the model," but I guess these predictions do help to capture structure semantics. That may also be the main reason for the poor performance of DOBF without MLM initialized. Besides, the authors state that "DOBF leverages the structural aspect of programming languages" in the abstract, but I can not see how DOBF leverages the structural aspect. Anyway, variables are more about ambiguous information instead of structure information.
>
> **Answer:**
> We agree with you on the fact that variables are more about ambiguous information than about structural information. When we wrote that DOBF leverages the structural aspect, we meant that we leverage the structure of programming languages to find these tokens and mask them appropriately (i.e. masking all the occurences of a variable so that the model cannot copy-paste the token). This contrasts with MLM applied to natural language where people usually mask words at random. We will clarify this in the paper.
>
> **Reviewer comment:**
> DOBF masks all occurrences of a variable with the same special token since the authors believe "it will be easy for the model to simply copy its name from one of the other occurrences." From my point of view, learning to copy which identifier to the masked position is a non-trivial task, which will help capture contextual information. However, masking with the same token will tell the model that the two variable is the same. I believe both masking strategies have their merits and they are somewhat complementary, but there is no in-depth evaluation to compare them. I see the overall performance improvement, but I am wondering whether this strategy works.
>
> **Answer:**
> DOBF and MLM/DAE are indeed complementary. We briefly mentioned it in the paper (lines 332-334, just before the conclusion), but we agree with you that some of the statements at the beginning of the paper were not clear enough. We clarified them in the updated version of the paper. Thank you for pointing this out.
>
> As to whether this strategy works, our experiments show that pre-training using DOBF initialized with MLM substantially outperforms MLM on downstream tasks. We also outperform our baseline pre-trained models (CodeBERT and GraphCodeBERT) on a variety of tasks when training with DOBF and the denoising objective at the same time.
>
>
> **Reviewer comment:**
> I am very interested in the experimental results of deobfuscation. According to a previous code summarization work [1], it is non-trivial to generate meaningful code summaries when removing all variable identifiers in a function. Though generating code summaries is different from recovering masked variables, it is surprising that DOBF_1 recovers 45.6% of the initial identiﬁer names, and the recovered code in Figure 2 looks perfect. How do you split the training set and test set for this evaluation? Maybe there is something that I misunderstand, but I guess the surprising performance may attribute to the memorization of duplicated code rather than the generalization abilities.
>
> **Answer:**
> This is a valid concern as code duplication can cause issues for tasks related to programming languages. We were also wary of code duplication and train-test contamination. Here are the steps we took to avoid it:
> - We removed all duplicate obfuscated and standardized files, so we don’t have any examples in the training set that have exactly the same structure as examples in the test set (the standardization standardizes the whitespaces, newlines and indentation and the obfuscation standardizes the identifier names).
> - We added files to the train, validation, and test splits according to the name of their repositories, so that files from the same repository and from forked repositories are all in the same split. It removes many of the near duplicates (slightly modified forked files). It also ensures that the model cannot learn the coding conventions of a project in the test set using data in the train set and generalizes to unseen repositories.
>
> Looking at our results on examples also convinced us that the good performance of our model is not attributable to memorization. In many instances, the model proposes some identifier names which are appropriate but different from the ground truth. For instance, the recovered code for BFS on Figure 2 may look perfect to us, but it was not perfect according to our metric. The second parameter VAR_1 (start node) is recovered as “start” but was “node” in the ground truth and the variable VAR_4 (popped element) is recovered as “node” but was “s” in the ground truth so the accuracy is only 71% for this example.
>
> The example on Figure 8 in the appendix may also be relevant. We provide a table with every ground truth and proposed identifier names and we can see that while the model predicts the less ambiguous names perfectly (e.g. “self” for the first parameter of each method, “forward” for the forward pass, “LSTM” for the name of the class”) it often proposes names that do not match the ground truth, but are still appropriate in the context (e.g. “init_weights” instead of “reset_parameters” or “prev_h” and “prev_c” instead of just “h” and “c”).
> We also tested DOBF on some unreleased code and found its output convincing.
>
> **Reviewer comment:**
> Regarding the experiments of the downstream tasks, the overall performance improvements over strong baselines (e.g., GraphCodeBERT) look like mainly coming from DAE, which is not the technical contribution of this paper. Without DAE, DOBF demonstrates no obvious superiority.
>
> **Answer:**
> The comparison with DAE trained from MLM without DOBF is indeed necessary to demonstrate the effectiveness of DOBF. Thank you for pointing this out. We added it as a baseline in our main result table (Table 2) and added references and discussions about this baseline in the text. Here is a link to the new results table: https://ibb.co/521RNkh.
>
> DAE beats MLM on most tasks (all but Java -> Python translation) but still underperforms DOBF + DAE on most tasks. Interestingly, DAE beats every other method for Python -> Java translation, providing small improvements compared to DOBF + DAE: +1.7% and +1.9% CA@1 with beam size 1 and 10 respectively. However, it underperforms every other pre-training method for Java -> Python, with CA@1 8.5% and 9.6% below those of DOBF + DAE with beam size 1 and 10. Overall, adding DOBF during training provides clear gains compared to training with DAE only for several tasks. For code translation, DOBF+DAE is only slightly below DAE for Python -> Java and clearly above for Java -> Python.
>
> **Reviewer comment:**
> Besides, there have been a pre-training method to predict a function name [2]. The idea is not quite new given the current technical trend, though it is a more general method targeting more downstream tasks of programming languages. [2] should be an important missing reference.
>
> **Answer:**
> Thank you, this recent paper is indeed relevant and we updated the paper to cite it.

---

### Author Response · Authors · 2021-08-09
**Modifications following the reviews**

We thank all the reviewers for the feedback and comments. We replied to each of them individually, and updated the paper following their suggestions.
- We ran additional experiments and added a baseline for DAE initialized with MLM and some discussions about the results.
- We made it clearer that MLM and DAE are complementary with DOBF and that both are useful to get good performances.
- We modified the argument about the societal impact of our work following the comments of Reviewer 3
- We made a few other modifications following the comments. You can find out more details in the answers to the reviewers.

---

### Decision · Program_Chairs · 2021-09-27

**Decision:**

Accept (Poster)

**Comment:**

The reviewers appreciated the key idea presented in the paper of using a new pre-training objective DOBF to recover obfuscated variable names from programs, and its usefulness for several downstream tasks. Adding a new DAE-only baseline for comparison was also greatly appreciated. While there were still some concerns regarding fairness of comparisons across models, overall the recommendation is for acceptance. Hopefully the authors can incorporate the detailed feedback from reviews and add additional experiments in the final version.